



# Biomass burning combustion efficiency observed from space using measurements of CO and NO₂ by TROPOMI

Ivar R. van der Velde[1,2], Guido R. van der Werf[1], Sander Houweling[1,2], Henk J. Eskes[3], J. Pepijn
Veefkind[3,4], Tobias Borsdorff[2], and Ilse Aben[1,2]

[1]Faculty of Science, VU University, Amsterdam, The Netherlands
[2]SRON Netherlands Institute for Space Research, Utrecht, The Netherlands
[3]KNMI Royal Netherlands Meteorological Institute, De Bilt, The Netherlands
[4]Department of Geoscience and Remote Sensing, Delft University of Technology, Delft, The Netherlands

*Correspondence to*: Ivar R. van der Velde (i.r.vander.velde@vu.nl)

**Abstract.** The global fire emission inventories depend on ground and airborne measurements of species-specific emission factors (EFs), which translate dry matter losses due to fires to actual trace gas and aerosol emissions. The EFs of nitrogen oxides ($NO_x$) and carbon monoxide (CO) can function as a proxy

for combustion efficiency to distinguish flaming from smoldering combustion. The uncertainties on these EFs remain large as they are limited by the spatial and temporal representativeness of the measurements. The global coverage of satellite observations has the advantage to fill this gap, making these measurements highly complementary to ground-based or airborne data. We present a new analysis of biomass burning pollutants using space-borne data to investigate the spatiotemporal efficiency of fire

combustion. Column measurements of nitrogen dioxide and carbon monoxide ($XNO_2$ and $XCO$) from the TROPOspheric Monitoring Instrument (TROPOMI) are used to quantify the relative atmospheric enhancements of these species over different fire-prone regions around the world. We find spatial and temporal patterns in the $\Delta XNO_2/\Delta XCO$ ratio that point to distinct differences in biomass burning behavior. Such differences are induced by the burning phase of the fire (e.g. high temperature flaming vs.

low temperature smoldering combustion) and burning practice (e.g. the combustion of logs, coarse woody

debris and soil organic matter vs. the combustion of fine fuels such as savanna grasses). The sampling

techniques and the signal-to-noise of the retrieved $\Delta XNO_2/\Delta XCO$ signals were quantified with WRF-

CHEM experiments and showed similar distinct differences in combustion types. The TROPOMI

measurements show that the fraction of surface smoldering combustion is much larger for the boreal forest

fires in the upper northern hemisphere and peatland fires in Indonesia. These types of fires cause a much

larger increase (3 to 6 times) in $\Delta XCO$ relative to $\Delta XNO_2$ than elsewhere in the world. The high spatial

and temporal resolution of TROPOMI also enables the detection of spatial gradients in combustion

efficiency at smaller regional scales. For instance, in the Amazon, we found higher combustion efficiency

(up to 3-fold) for savanna fires than for the nearby tropical deforestation fires. Out of two investigated

fire emission products, the TROPOMI measurements support the broad spatial pattern of combustion

efficiency rooted in GFED4s. Meanwhile, TROPOMI data also add new insights on regional variability

in combustion characteristics that are not well represented in the different emission inventories, which

can help the fire modeling community to improve their representation of the spatiotemporal variability in

EFs.

## 1 Introduction

The importance of biomass burning as a source of atmospheric trace gases and aerosols has been

increasingly studied and recognized in the past decades (Andreae, 2019). To quantitatively assess the

influence of biomass burning on atmospheric chemistry and climate the atmospheric modeling

community requires accurate estimates of fire emissions. Important scientific efforts have led to the

development of a number of biomass burning emission products by combining satellite derived datasets

of burned area with biogeochemical models and biomass density datasets that enabled more accurate

emission estimates (e.g. Hoelzemann et al. 2004; Ito and Penner, 2004; van der Werf et al., 2003). Since

then much progress has been made to reduce uncertainties of the involved datasets (e.g. burned area, fuel

loads, combustion factors, and emission factors), but the uncertainties in the emission estimates remain

substantial especially at the more detailed regional scales (van der Werf et al., 2017). The recent

emergence of new space-based instruments that measure different trace gases could provide additional

top-down constraints on biomass burning emissions and combustion characteristics.

Since the 1980s numerous field measurement campaigns have provided information on biomass burning

characteristics and emissions for different biomes and vegetation types around the world (e.g. Andreae et

al., 1988; Lacaux et al., 1996; Yokelson et al., 1999). Most of these studies derived so-called emission

factors (EF or EFs) for different chemical compounds to quantify the number of grams of a trace gas or

aerosol emitted per kilogram of biomass burned. These EFs are combined with biogeochemical models

such as used in the Global Fire Emissions Database (GFED; van der Werf et al., 2010) to provide global

biomass burning emission estimates, which in turn are used as input for atmospheric transport models

(e.g. CarbonTracker data-assimilation system; Peters et al., 2007). The main function of these

biogeochemical models is to help predict the spatiotemporal combustion rate of biomass dry matter based

on the fuel load, combustion completeness and / or remotely-sensed products like burned area or fire

radiative power (FRP). A number of EF databases have been published providing biome-average EFs

derived from the large collection of available field and laboratory measurements. The first widely used EF database came from Andreae and Merlet (2001), followed by Akagi et al. (2011), who introduced additional biome categories and selected only measurements of fresh smoke plumes, before significant photochemical processes occurred. The latter improved the consistency with atmospheric transport

models that use fire emissions as direct inputs before the internal chemistry parameterizations affect the emitted tracers. However, these databases do not account for the variability in EFs within the same biome, which can be substantial and introduces a major source of uncertainty (van Leeuwen et al., 2013). Natural variations in the chemical and structural composition of biomass, temperature, moisture content, and wind speed can cause large variations in the relative fraction of flaming and smoldering combustion. As a

consequence, actual EFs may vary substantially calling for more detailed information to move beyond the use of biome average values.

The lack of spatial representativeness in EF estimates can partly be resolved by increasing the field measurement effort. In addition, key information on biomass burning characteristics can be retrieved from

space-based instruments, as it is reflected in the atmospheric composition of different trace gases. The main advantage of these instruments is the large spatial and temporal coverage that can be achieved, compensating limitations in spatial resolution and surface sensitivity. Therefore, satellite measurements of regional trace gas enhancements have the potential to provide valuable information on combustion efficiency, burning practices, fuel type and their variability, in particular in remote areas where we lack

ground-based measurements and other detailed information. Two trace gases of particular interest are commonly measured from space: carbon monoxide (CO) and nitrogen dioxide ($NO_2$). Enhanced

atmospheric abundances of these two species due to fires provide a unique atmospheric fingerprint of biomass combustion efficiency, i.e., the fraction of biomass combustion by flaming and smoldering. Flaming combustion is hotter and cleaner and produces relatively large amounts of $NO_2$ and relatively

small amounts of CO, whereas smoldering combustion happens at fairly low temperatures at the surface and produces predominantly CO (Andreae and Merlet, 2001).

Previous applications of joint trace gas analysis, including CO and $NO_2$, focused mostly on constraining anthropogenic and fossil fuel emissions, using either surface observations (e.g. Lopez et al., 2013; Hassler

et al., 2016) or satellites (e.g. Silva et al., 2013; Reuter et al., 2014; Konovalov et al., 2016). Mebust and Cohen (2013) demonstrated the detection of seasonal variations of fire EFs in the African savannas using satellite measurements of $NO_2$. Silva and Arellano (2017) used satellite observations of CO, $CO_2$ and $NO_2$ in a novel way to distinguish combustion types around the world. This study provided new insights on emission inventories as they found distinct differences in the ratios of $CO/CO_2$ and $CO/NO_2$ between

different biomass and urban combustion regions, which are often not well represented in emission inventories.

In this study we aim to demonstrate the capabilities of the new space-borne TROPOspheric Monitoring Instrument (TROPOMI, launched in October 2017; Veefkind et al., 2012) to provide new information

about biomass burning characteristics and efficiency in different regions around the globe. The main advantage of TROPOMI is that it delivers co-located column densities of several trace gases, including CO and $NO_2$. It extends the capability of legacy instruments like MOPITT and OMI by measuring trace



gases at improved accuracy, surface sensitivity, and spatial resolution providing daily global coverage. The wealth of data that TROPOMI provides offers the unique opportunity to monitor seasonal changes

in the relative amount of flaming and smoldering combustion, even in remote regions where ground-based measurements of fire properties are sparse. For instance, under relatively clear-sky conditions, the Amazon basin can now be examined for trace gases in much more detail on a day-to-day basis during annual dry season fire spells. TROPOMI surmounts some of the limitations of previous joint trace gas analysis studies where measurements were often taken from various instruments, each with their own

intrinsic limitations such as a clear-sky-only retrieval requirement (e.g. with MOPITT), and with widely different spatial resolutions and repeat cycles, i.e. the number of days between two satellite overpasses over the same region. The improved consistency among the different TROPOMI data products in terms of overpass time and location, retrieval sensitivity and spatiotemporal resolution might also help to suppress aggregation errors and biases in the derived ratios of trace gases, improving the capability to

distinguish differences in combustion types at the regional scale. The aim of this study is two-fold: (1) To demonstrate the detection of spatial variations in the regional enhancements of CO and $NO_2$ for different fire prone areas that are either dominated by smoldering or flaming fires, or a combination of both. (2) To investigate the use of TROPOMI CO and $NO_2$ to verify the current set of biome-specific EFs used in the atmospheric and climate modeling community.



## 2 Methodology

### 2.1 GFED4s and GFAS Emission Factor Ratio

We used two well-established biomass burning emission datasets to interpret and validate TROPOMI inferred combustion characteristics and efficiencies: the Global Fire Emission Database version 4 with small fires (GFED4s; van der Werf et al., 2017) and the Global Fire Assimilation System version 1 (GFAS; Kaiser et al., 2012). Both datasets provide global fire emission fluxes for a large number of chemical species but use different methods.

GFED4s is based on Carnegie–Ames–Stanford Approach (CASA) biogeochemical model (Potter et al., 1993) to predict the amount of above and below ground biomass at monthly temporal resolution. The MODIS Collection 5.1 MCD64A1 500 m burned area satellite product (Giglio et al., 2013) is used to estimate the daily dry matter combustion rate at 0.25°×0.25° spatial resolution from 2001 up to 2016. GFED4s also includes 1×1 km$^2$ thermal anomalies (active fire counts) from Terra and Aqua MODIS, and 500×500 m$^2$ surface reflectance observations, providing a statistical estimate of the burned area associated with small fires (Randerson et al., 2012; van der Werf et al., 2017). The GFED4s flux estimates from 2017 onward (used in this study) are not directly derived from the burned area product because the underlying MODIS algorithm was upgraded from Collection 5.1 to Collection 6. Instead, flux estimates are simply derived from MODIS active fire detections and their FRP and the climatological ratio between them derived from the overlapping 2003-2016 period. The GFAS product calculates emissions by assimilating FRP observations from the MODIS Terra and Aqua satellites and is tuned to match the dry



matter combustion rate of GFED3 per biome (Kaiser et al., 2012). The version we used provides daily

emissions at 0.1°×0.1° spatial resolution.

Both biomass burning products are combined with EFs to translate the derived dry matter combustion

rate to specific trace gas and aerosol emissions. These EFs are based on a large number of trace gas

measurement campaigns in the field, in the air or in the laboratory, and are subdivided for dominant

biome/burning categories without specifying any variability in space and time. GFAS uses the older EF

dataset compiled by Andreae and Merlet (2001) with additional updates from the literature and EFs of

peatland fires from Christian et al. (2003). In this dataset boreal and temperate forest fires form together

a single category named extratropical forest fires (ETF). GFED4s uses EFs largely based on the dataset

compiled by Akagi et al. (2011). This dataset is based on trace gas measurements from fresh smoke

sampled in close proximity of the fire source and cooled to ambient temperature but with minimal

photochemical processing. This provides a better representation of the initial emissions without chemical

disturbances (to aid assessment of biomass burning in atmospheric chemistry models). The Akagi et al.

(2011) dataset makes a distinction between boreal and temperate forest fires. For boreal fires they used

the average of airborne and ground-based measurements that is roughly equivalent of assuming 70% of

dry matter consumption is originating from smoldering combustion. Therefore, the EFs for the boreal

latitudes are relatively high for carbon monoxide ($EF_{CO}$: 127.0 g kg$^{-1}$) and low for nitrogen oxides ($EF_{NOx}$:

0.90 g kg$^{-1}$). The EFs for the temperate fires are 88.0 and 1.92 g kg$^{-1}$, respectively for CO and NO$_x$, and

represent a larger fraction of flaming combustion similar to the ETF category used in the Andreae and

Merlet (2001) dataset. Other variations in EFs between Akagi et al. (2011) and Andreae and Merlet (2001)

are due to variations in the averaging and weighting methods of the measurements. In addition, GFED4s

includes sub-grid cell partitioning of burned area to account for different fire types within a grid cell,

which affects the grid-average emissions of CO and $NO_x$. Because NO is usually the most abundant N-

species emitted to the atmosphere and because NO and $NO_2$ are rapidly interconverted in the atmosphere

both datasets report EFs for $NO_x$ as NO. Henceforth, the EFs are reported in units of mmol kg$^{-1}$ and mol

kg$^{-1}$ for $EF_{NOx}$ and $EF_{CO}$, respectively, to make ratios of EFs of similar magnitude as the ratios of column

densities measured by TROPOMI (see Sect. 2.2). Table 1 shows an overview of EFs of CO and $NO_x$ used

by GFAS and GFED4s.

The spatial stratification of the different biomass burning categories is apparent in the ratio between $NO_x$

and CO EFs. This ratio exhibits a distinct "fingerprint" that carries information on combustion efficiency,

combustion practice, and fuel type. In this study, we call this dimensionless metric the Emission Factor

Ratio (EFR = $EF_{NOx}/EF_{CO}$). EFR is a relative measure of how much millimoles of $NO_x$ are released to the

atmosphere for each mole of CO. This metric is a proxy for the modified combustion efficiency (MCE)

parameter that is often used in fire emission quantification studies but more difficult to derive from space

given the relatively small departures of $CO_2$ concentrations over biomass burning regions from

background conditions. The MCE is defined as $\Delta CO_2/(\Delta CO+\Delta CO_2)$ to indicate combustion efficiency of

a fire by measuring the amount of excess in $CO_2$ in comparison to total emitted C from $CO_2$ and CO

(Yokelson et al, 1999). Table 1 gives the EFR for the different combustion types based on the ratio

between the EFs of $NO_x$ and CO used by GFED4s and GFAS. Figures 1a and 1b show the spatial

distribution of EFR in both datasets. For GFED4s, we subdivided EFR into three different categories:

high EFR above 50 for savanna fires, EFR between 10 and 50 for temperate forest fires, tropical

deforestation fires and agricultural waste burning, and EFR lower than 10 for boreal and peatland fires.

High EFR is thus related to the flaming type of combustion that is hotter and more efficient as it produces

relatively less CO alongside $CO_2$ and relatively more $NO_x$ by combustion of N in the biomass itself.

Conversely, low EFR is generally related to slow smoldering type of combustion. The EFR categories are

similar for GFAS, however, due to differences between the EFs datasets (for reasons discussed in the

previous paragraph) EFRs are classified differently: high EFR above 30 for savanna fires, EFR between

10 and 30 for ETF fires, tropical deforestation fires and agricultural waste burning, and EFR lower than

10 for peatland fires.

Highlighted in Fig. 1 are various regions of interest studied in this paper with strong seasonal occurrences

of biomass burning. Regions that have been selected for detailed analysis using TROPOMI: two boreal

fire regions in North America, one boreal fire region in Siberia, five savanna fire regions on the African

continent, one savanna fire region in Australia, two peatland fire regions in Indonesia, and 15 regions in

South America to more specifically study spatial gradients in combustion efficiency between tropical

deforestation and savanna fires.

### 2.2 TROPOMI CO and NO₂

The TROPOMI instrument was launched on 13 October 2017 onboard the Sentinel-5 Precursor satellite

to monitor the chemical composition of the atmosphere (Veefkind et al., 2012). It measures a range of

trace gases at unprecedented spatial resolution with a daily global coverage. Sect. 2.2.1 and 2.2.2 provide

further details about the TROPOMI operational level 2 column density data products of carbon monoxide

(XCO) and nitrogen dioxide (XNO$_2$). Figure 2 shows a few examples of monthly and daily average maps

of XCO and XNO$_2$ for a number of biomass burning regions together with CO emissions from GFED4s

(Sect. 2.1). Enhancements in XCO and XNO$_2$ correspond well with local fire emissions based on

independently derived burned area. Note that the chemical lifetime of NO$_x$ is much shorter than for CO

(minutes to hours vs. weeks to months). The main chemical driver during daytime is the photochemical

balance between NO$_2$ photolysis and NO oxidation by ozone converting NO into NO$_2$ and makes NO$_2$ a

robust measure for NO$_x$. The NO$_x$ lifetime is limited by the conversion of NO$_2$ to HNO$_3$ in reaction with

hydroxyl (OH) radicals. The short chemical lifetime results in a precise alignment between the

enhancements of XNO$_2$ and the location of fire emissions, while enhancements of XCO are more affected

by atmospheric transport due to its longer chemical lifetime. These differences in lifetime can cause biases

in the joint analysis of XNO$_2$ and XCO and its ratio. That limits our ability to make direct quantitative

comparisons between EFs and column densities. Nonetheless, assuming the lifetime does not vary greatly

from fire to fire and from region to region, it is probable that it does not affect our ability to the detect

variations in fire characteristics around the world. This limitation is further discussed in Sect. 4 of the

paper.

### 2.2.1 XCO

The carbon monoxide total column density from TROPOMI is retrieved from reflected and backscattered

solar radiance around 2.3 μm measured by the shortwave infrared module of the spectrometer. The

Shortwave Infrared Carbon Monoxide Retrieval algorithm (SICOR, Landgraf et al., 2016) is used to

translate spectral radiances to XCO column densities, with high sensitivity to the planetary boundary layer

for clear-sky conditions over land. For cloudy conditions over land and ocean, the XCO has a stronger

sensitivity at higher altitude. To account for cloud interferences SICOR retrieves an effective cloud optical

depth and cloud height, and provides a column averaging kernel as part of the product which represents

the height sensitivity of the measurement.

A good agreement was found between TROPOMI XCO and TCCON XCO ground measurements for

clear and cloudy sky conditions (Borsdorff et al., 2018a). Mean biases amount to: 6.0 ppb for clear-sky,

6.2 ppb for cloudy-sky retrievals and 5.8 ppb for the combination of both. The station-to-station standard

deviation of the bias was 3.9 ppb for clear-sky, 2.4 ppb for cloud-sky, and 2.9 ppb for the combination of

both. Thereby, TROPOMI achieves its mission requirements on precision (<10%).

The XCO column density for 2018 is observed with daily global coverage at a spatial resolution of 7×7

km$^2$ in nadir. The data is selected for clear-sky and cloudy-sky conditions with a cloud top height limited

to 5000 m and an aerosol optical thickness equal or larger than 0.5 (TROPOMI CO level 2 README

Document, 2018). In addition, the two most westward pixels of the swath were excluded due to

performance issues (Borsdorff et al., 2018b). The XCO column density is presented in units of mole per

square meter (mol m$^{-2}$).

**2.2.2 XNO$_2$**

The tropospheric nitrogen dioxide column density from TROPOMI is retrieved from spectrometer

measurements of direct and backscattered solar radiance between 405 and 465 nm. The XNO$_2$ retrieval

algorithm uses the DOAS approach and is an adapted version of the algorithm used for the DOMINO

v2.0 (Boersma et al., 2011) and QA4ECV XNO$_2$ products (van Geffen et al., 2015, Boersma et al., 2018).

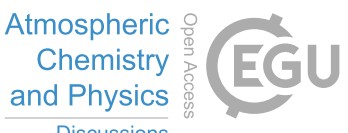

In the retrieval procedure, $NO_2$ slant columns are derived from the measured spectra using the DOAS method. Then the tropospheric component of the slant columns is separated from the stratospheric component, and finally the tropospheric slant columns are converted to vertical columns $XNO_2$ based on the tropospheric air mass factor (AMF).

$XNO_2$ is observed with daily global coverage at a spatial resolution of $3.5 \times 7$ $km^2$ in nadir. The spectrometer's near infrared band provides additional information on cloud characteristics and allows a better cloud correction, i.e., improving the measurement precision under cloudy conditions. $XNO_2$ column densities have been compared with ground-based MAX-DOAS measurements at 14 stations. In general, TROPOMI underestimates the tropospheric column at polluted sites. The daily median negative

biases are generally less than 50%, within the required measurement precision, but vary from station-to-station (TROPOMI $NO_2$ level 2 README Document, 2018). Because this bias is largely systematic, it is not expected to deteriorate our ability to differentiate between fire characteristics. This is further investigated in Sect. 3.1.

For this study, we use the recommended filter settings outlined in the README document, removing cloud-covered scenes with a cloud radiance fraction exceeding 0.5, scenes covered by snow or ice, and other problematic retrievals (qa_value > 0.75). The $XNO_2$ column density is presented in units of millimole per square meter ($mmol\ m^{-2}$).

### 2.3 Mole Fraction Ratio: sampling methods

For our analysis, we selected important hotspots of biomass burning according to the GFED4s database. To prevent contamination with urban trace gas emissions large population centers were avoided. The regions are outlined in Fig. 1 and 2. Within each region we collected all the available XCO and XNO$_2$ data that passed the filters explained in Sect. 2.2.1 and 2.2.2 for up to three consecutive months depending on the timing and duration of the fire season.


To derive the regional enhancements in XCO and XNO$_2$ relative to the background, $\Delta$XCO and $\Delta$XNO$_2$ respectively, we used two main sampling methods depending on the scale and severity of the fires in the region. A statistical bulk method (SBM) is used for regions that are characterized by extensive seasonal burning over a large area and where upwind background levels of XCO and XNO$_2$ are difficult to define.

A local sampling method (LSM) is used for regions where we could identify local fires and plumes of enhanced trace gas abundance for which the wind direction and background column density upwind of the fires could be determined. Each method is discussed in more detail in Sect. 2.3.1 and 2.3.2, respectively. With daily estimates for $\Delta$XNO$_2$ and $\Delta$XCO we were able to derive a new dimensionless metric: the Mole Density Ratio (MDR = $\Delta$XNO$_2$/$\Delta$XCO). The MDR is the atmospheric equivalent to EFR

and provides a remotely sensed proxy for biomass combustion efficiency.

### 2.3.1 Statistical bulk method

The statistical bulk method (SBM) is based on the method discussed in Silva and Arellano (2017) who used it to distinguish urban and industrial trace gas enhancements from biomass burning. It provides a simple measure of regional trace gas enhancements when background column densities are difficult to





determine. For this method, the daily TROPOMI data were regridded at 0.1°×0.1° resolution from which

co-located XCO and XNO2 data within 5°×5° boxes were sampled each day over the selected regions

(see Fig. 1 and 2). The size of these boxes allows for a sufficient number of trace gas observations each

day (more than 1000). To determine the regional trace gas enhancement relative to the background

($\Delta$XCO and $\Delta$XNO2 are here jointly indicated by $\Delta X$) we assume that the sampled data exhibits a Gaussian

normal distribution. A trace gas enhancement of one standard deviation above the daily mean of the

distribution is assumed to be due to fires, i.e., $X_{fire} = \mu_X + \sigma_X$. Conversely, a column density of one standard

deviation below the mean is assumed to represent the trace gas background, i.e., $X_{BG} = \mu_X - \sigma_X$. This

implies that the regional trace gas enhancement is assumed to be two times the standard deviation of the

distribution, i.e., $\Delta X = X_{fire} - X_{BG} = 2\sigma_X$. Figure 3a displays an idealized normal distribution of sampled

column densities of XCO indicating the values of $XCO_{fire}$, $XCO_{BG}$, $\Delta XCO$, $\mu_{XCO}$, and $\sigma_{XCO}$ along the

distribution. The MDR between $\Delta$XNO2 and $\Delta$XCO is therefore equal to the ratio between two standard

deviations $\sigma_{XNO2}/\sigma_{XCO}$. As discussed by Silva and Arellano (2017), this assumption is only valid if both

species are highly correlated with each other. This is the case for this study given the strong co-location

of the sampled XCO and XNO2 data, the daily sampling interval for both species, and because we

carefully selected strong biomass burning source regions. SBM was used for the following regions: 15

regions over the southern Amazon basin of South America, where data was sampled between July and

September 2018, 2 regions over Northern Africa, where data was sampled in December 2018, and 3

regions over Southern Africa, where data was sampled between July and September 2018. Deriving $\Delta X$

as outlined above, may not reflect a formally correct estimate of the regional trace gas enhancement

relative to the actual background, but that is also not our main goal. The purpose is to have a consistent



method among the two trace gases that provides a reasonable proxy for regional fire induced column

enhancements. Therefore, this method was only used for regions where we have a very high density of

fires within our study area and where it is difficult to investigate individual fire plumes and their

background mole density levels. Some of the errors introduced by this method are systematic and have a

similar impact on $\Delta XNO_2$ and $\Delta XCO$ (e.g. error due to atmospheric transport) and will cancel out in the

estimate of MDR. Other errors may introduce new uncertainties and biases on top of the TROPOMI

column uncertainty unevenly between $XNO_2$ and $XCO$, potentially affecting our ability to differentiate

between combustion characteristics. For instance, the assumption of a Gaussian normal distribution of

the sampled data might not hold for one or both of the trace gases. To assess the importance of these

uncertainties, we developed two alternative methods to derive $\Delta X$ that are closely related to SBM. The

first alternative method (SBM_alt1) assumes that $\Delta X$ is not determined by the standard deviation but by

the difference between the 15.9 and 84.1 percentile ranks around the median of the distribution. Figure

3b shows an example of such a distribution. Only if the sampled data is perfectly normally distributed,

SBM_alt1 and SBM will yield the same result because the two percentile ranks will align with the minus

one and plus one standard deviations. Variations from the standard normal could for instance deteriorate

our ability to differentiate between combustion characteristics as it will affect the estimates for $\Delta X$ and

MDR. The second alternative method (SBM_alt2) derives $\Delta X$ by taking the difference between $X_{fire}$ from

the standard SBM method and an alternative $X_{BG}$ derived from a distribution of samples from an adjacent

$5° \times 5°$ region. Naturally that means the SBM_alt2 $\Delta X$ value is only identical to the standard SBM $\Delta X$

value if both estimates for $X_{BG}$ are identical. Figure 3c shows an example of this method with an idealized

background and source distribution of sampled XCO.

SBM and the two alternative methods have been validated for two source regions in South America (see Sect. 3.1). The first region is located south of the Amazon river over the Brazilian state of Amazonas in

the tropical rainforest and is dominated by deforestation fires, i.e. the practice of burning logs and debris that remain on the landscape after initial clearing to create new agricultural land. The second region is located over the central Brazilian state of Goias in an ecoregion called the Cerrado, which is a savanna-like fire-adapted ecosystem with frequent fires that mostly consume the grass layer but where the expansion of agriculture is also an important cause of fires. These two areas are shown in Fig. 4, in green

and blue, respectively. In the 3-month dry season between July-September 2018, the parameters $X_{fire}$, $X_{BG}$, $\Delta X$ were determined every day for XCO and XNO$_2$ using SBM, SBM_alt1, and SBM_alt2. The latter method used the two adjacent background regions shown in purple in Fig. 4. These two background regions were chosen for a number of reasons. First of all, the background region for the Cerrado savanna fires was on average upwind of the source area. The average wind direction in the planetary boundary

layer of the domain was predominantly from the east during the 3-month period (see Fig. 4), based on a WRF-CHEM simulation nudged to NCEP re-analysis boundary conditions. Moreover, the CO and NO$_x$ emissions from fires were about two times smaller in the background region than in the source region according to the GFED4s database (0.6 GgCO region$^{-1}$ day$^{-1}$ vs. 1.1 GgCO region$^{-1}$ day$^{-1}$). Similarly, we opted for a 'clean air' background area just northeast of the deforestation region where CO and NO$_x$

emissions from fires were very small during our study period.


### 2.3.2 Local sampling method

The local sampling method (LSM) is a more straightforward approach to determine local enhancements in trace gas densities in close proximity of the actual fire hotspot. This method was specifically used for fires in the North American boreal biome in July 2018, the Siberian boreal biome in July and August 2018, the central Australian savanna biome in November and December 2018, and the Indonesian peatland biome in August and September 2018 (see Fig. 1 and 2). All events were relatively isolated from other fires, a prerequisite for using this method. For predefined 5°×5° and 10°×10° boxes, TROPOMI data were regridded at 0.1°×0.1° resolution. Subsequently, co-located XCO and $XNO_2$ data each day are sampled within a radius of 10 km from a location where CO and $NO_x$ was emitted according to GFED4s. For each fire hotspot, $X_{fire}$ is defined as the average of these sampled column densities. The background column density $X_{BG}$ is determined each day by taking the average of all sampled column densities inside a smaller subregion upwind of the fire hotspot within the larger predefined box. The location of the background subregion was determined by visual inspection, looking at the predominant direction of the individual trace gas plumes. For each day, we averaged $\Delta XCO$ and $\Delta XNO_2$ over active hotspots in the predefined boxes, which were subsequently used to derive a daily average MDR. Days with insufficient data upwind of the fire hotspots were excluded from the analysis as well as days with enhanced trace gas levels that were advected into the region from outside. For instance, we had to filter out by visual inspection a number of days for the North American regions because high amounts of CO were advected from the Eurasian continent to Alaska obscuring most of the local enhancements in CO.



## 2.4 WRF-CHEM


To evaluate the methodology of the joint analysis of TROPOMI XCO and $XNO_2$, and in particular the SBM sampling technique, we used the Weather Research Forecasting model version 4.0 coupled with chemistry (WRF-CHEM). Main purpose was to investigate whether the sampling techniques can provide estimates of XCO, $XNO_2$ and MDR that are distinctly different between four combustion types. Synthetic

WRF-CHEM simulations were performed using a single domain located over the northern part of South America stretching over 6000 km in the east-west direction and 3900 km in the north-south direction (see Fig. 4). We used a horizontal resolution of $30\times30$ km$^2$ with 32 vertical levels. We chose the 'tropical' suite of physics options that includes Yonsei University (YSU) scheme for planetary boundary layer physics (Hu et al., 2013), WSM 6-class scheme for microphysics (Hong and Lim, 2006), Tiedtke-scheme

for cloud physics (Tiedtke, 1989), and Rapid Radiative Transfer Method (RRTM, Mlawer et al., 1997) for short- and longwave radiation. We included gas-phase chemistry mechanisms from the Regional Acid Deposition Model version 2 (Stockwell et al., 1990), but without aerosol chemistry. Boundary and input meteorological fields for September 2018 came from NCEP FNL Operational Global Analysis dataset prepared on a $1°\times1°$ grid every 6 hours. Boundary and initial trace gas concentrations were taken from

CAMS model and were interpolated to WRF vertical levels. GFED4s (see Sect. 2.1) provided the biomass burning dry matter combustion rate based on real fire events for South America in September 2018. These were multiplied with a set of EFs to acquire a synthetic estimate of biomass burning emissions for the entire domain that is associated with a single combustion type: either boreal fires, mixed peat fires, tropical deforestation fires or savanna fires. For the EFs of the mixed peat fires we assumed 60% is

combusted by tropical deforestation (to mimic overstory consumption) and 40% is combusted by peat



soils. As a consequence, each of these four emission estimates carried a different (but in space constant) EFR between $NO_x$ and CO over the entire domain for each of the respective fire types.

WRF-CHEM was executed four times for September 2018 under the exact same conditions and settings
described above, except that we used for each run one of the four modified biomass burning emissions. Each simulation provided hourly 3-dimensional fields of CO and $NO_2$ concentrations for the entire month. Close to the center of the domain we collected within a 5°×5° region each day at 2pm local time (half an hour later than the actual TROPOMI overpass) all CO and $NO_2$ data. These were translated into XCO and $XNO_2$ column mole densities using a daily mean estimate of TROPOMI's averaging kernel (AK) of the
two respective species, derived from September 2018 data over the same 5°×5° collection region. This assured realistic differences in column sensitivity for simulated XCO and $XNO_2$, even under cloudy conditions. The collection of column densities was used to derive daily $\Delta XCO$, $\Delta XNO_2$, and MDR using the three SBM sampling methods discussed in Sect. 2.3.1.

## 3 Results

In Sect. 3.1 we present a detailed analysis of XCO and $XNO_2$ data focusing on the Amazon basin. This analysis includes the errors associated with the different sampling methods, the significance of the retrieved MDR signatures in relation to the instrument precision of TROPOMI, and a comparison of retrieved MDR signatures between TROPOMI and two biomass burning datasets. In Sect. 3.2 we will discuss the MDR of other regions around the world. In Sect. 3.3 we provide an overview of all retrieved
MDR signatures and compare them with the regional patterns in EFR and with WRF-CHEM simulations.



### 3.1 South American deforestation and savanna fire characteristics

The joint analysis of $\Delta XCO$ and $\Delta XNO_2$ column densities in Fig. 5 shows a clear distinction between the deforestation and savanna regions during the 3-month dry season. For the deforestation burning region we observe much larger increased levels of $\Delta XCO$ (up to 0.030 mol m$^{-2}$) relative to $\Delta XNO_2$ mostly in

September indicating a substantial fraction of smoldering combustion later in the season. This is consistent with ground-based measurements of tropical forest fires that usually show a persistent smoldering phase that can continue for days, in particular when woody debris is ignited that is piled together (Carvalho et al., 2001; Morton et al., 2008). The day-to-day variability in $\Delta XCO$ is large but remains quite proportional to the variability in $\Delta XNO_2$ in September, indicating that the relative amount

of smoldering and flaming combustion remains relatively constant throughout the month. For the deforestation region $\Delta XCO$ and $\Delta XNO_2$ are correlated in September with a Pearson correlation coefficient of r=0.82, r=0.84, and r=0.68 for respectively SBM, SBM_alt1 and SBM_alt2. The estimates of $\Delta XNO_2$ are quite similar for the deforestation and the savanna region (between 0.005-0.05 mmol m$^{-2}$). However, lower $\Delta XCO$ values (up to 0.010 mol m$^{-2}$) point to a much cleaner combustion of savanna

biomass, which is common for savanna fires where the flaming phase typically dominates (Andreae and Merlet, 2001). In contrast to deforestation fires, $\Delta XCO$ and $\Delta XNO_2$ for savanna fires are less correlated (r<0.66), i.e. the trace gases do not change consistently on a day-to-day time scale.

In September, at the peak of the fire season, the monthly average MDR is significantly different for the

two fire types, irrespective of the sampling method used (see Fig. 5c). The MDR estimates for deforestation fires range between 1.06 and 1.55 whereas the MDR estimates for savanna fires are higher,

ranging between 2.77 and 3.17. The separation between deforestation and savanna fires remains also quite robust to the different bulk sampling methods used (SBM, SBM_alt1 or SBM_alt2). For each fire type, the monthly average $\Delta$XCO and $\Delta$XNO$_2$ estimates of the three sampling methods lay well within the 1$\sigma$

uncertainty level of each method (the standard deviation of day-to-day variability in $\Delta$XCO and $\Delta$XNO$_2$). The differences in MDR between SBM and SBM_alt1 are quite small in August and September (within 15% for both fire types) indicating that SBM is not so sensitive to a possibly non-Gaussian shape of the distribution. Much larger differences in MDR exist between SBM and SBM_alt2 (up to 35% for deforestation fires), with much larger day-to-day variability in MDR using SBM_alt2. It indicates the

background estimates of XCO and XNO$_2$ in the adjacent regions are not necessarily consistent with the background densities derived with the Gaussian fit. These uncertainties can be attributed to the incorrect assumption that the sampled trace gases are Gaussian distributed or by the somewhat arbitrary choice of the background regions for SBM_alt2, which do not necessarily characterize the true background each day in September. As discussed in Sect. 2.3.1, it is not our goal to determine the best possible background

estimate, which is difficult for these kinds of regions that are continuously surrounded by fires. Instead, we opted for a mathematical method that is consistent in application for both trace gases and provides a reasonable proxy for regional fire induced trace gas enhancements.

The influence of instrument precision on the $\Delta$XCO, $\Delta$XNO$_2$ and MDR estimates was also quantified.

The instrument precision of XCO under relatively clear-sky conditions is primarily a function of surface albedo (Landgraf et al., 2016). The low albedo of the tropical deforestation region yields an average XCO precision of 0.0014 mol m$^{-2}$, which is two times larger than for the savanna region with an average XCO

precision of 0.0007 mol m$^{-2}$. The precision of $XNO_2$ is around 0.011 mmol m$^{-2}$ for both regions and is dominated by air mass factor uncertainties under polluted conditions (Lorente et al., 2017). The

contribution of instrument/retrieval precision to the uncertainty in $\Delta XCO$, $\Delta XNO_2$ and MDR was estimated using a synthetic distribution of daily measurements for both trace gases in September 2018. This was done as follows: assuming no systematic errors or biases, each single TROPOMI measurement was randomly perturbed by its precision value or decreased by its precision value or remained unchanged. Subsequently, the SBM method was applied on this distribution, yielding an uncertainty range around

$\Delta XCO$, $\Delta XNO_2$ and MDR values due to instrument or retrieval noise (see Fig. 6). While the $XNO_2$ uncertainty is quite substantial in comparison to XCO, their contribution to $\Delta XNO_2$, and in the resulting MDR, is actually quite small on a monthly time scale. There is barely any overlap in the range of monthly MDR estimates of the deforestation and savanna fires, which means that differences in combustion characteristics easily exceed the precision. Only for smaller $\Delta XCO$ and $\Delta XNO_2$ values, TROPOMI's

precision will become a more limiting factor in terms of signal-to-noise and could explain the lower correlation between daily $\Delta XCO$ and $\Delta XNO_2$ noted before for savanna fires.

The September daily estimates of $\Delta XCO$ and $\Delta XNO_2$ are shown in Fig. 7a for the deforestation region, along with the smoothed estimates of the MDR, and the smoothed GFED4s and GFAS CO fire emissions.

In September, when the enhancements of the two trace gases and the CO emissions are at their maximum, the MDR is also at its lowest level around 1.5, indicating persistent less efficient combustion in the region (relatively less $NO_2$ and more CO release to the atmosphere). The same set of daily estimates are also shown for the savanna region in Fig. 7b. The peaks in $\Delta XCO$ and $\Delta XNO_2$ correspond with peaks in the

GFAS CO emissions and the MDR is quite constant between 3.0 and 3.5, i.e. about twice as high as the

MDR for the deforestation region.

It is worthwhile to note that the GFAS and GFED4s emission products do not necessarily align well in

Fig. 7a and 7b. As mentioned in Sect. 2.1, these two products use different methods and data products to

derive fire emission estimates. The difference between GFAS and GFED4s reflects the uncertainty in the

amount of CO emitted. This is apparent for the savanna region, where the daily GFAS CO emissions are

on average a factor of four larger than the GFED4s emissions. The 2018, GFED4s estimates are derived

using active fire detections and their FRP and a simple relationship based on the climatological FRP and

GFED4s ratio based on the 2003-2016 period for each $0.25° \times 0.25°$ grid cell. GFAS, on the other hand,

used the FRP associated with those active fires and is tuned to match GFED3 emissions but not for each

grid cell but for each biome (Kaiser et al., 2012). In addition, the used EFs between the two products are

different (see Table 1). For the large deforestation fires the estimates of CO emissions from GFAS and

GFED4s are more similar although the timing is somewhat different (see Fig. 7a).

We also demonstrate the detection of subcontinental scale gradients in biomass burning efficiency from

space. The September average MDR (derived with the standard SBM) is shown in Fig. 8a for 15 $5° \times 5°$

regions south of the Amazon river that cover the two main biomes: tropical rainforest (dominated by

deforestation fires) and Cerrado savanna (dominated by savanna fires). For the same 15 regions we also

derived the EFR from the monthly average GFED4s $NO_x$ and CO fire emissions shown in Fig. 8b. These

emissions are essentially based on the Akagi et al. (2011) EF database that is compiled from ground and



airborne measurements. Although the relationship is not perfect, MDR responds fairly linearly to EFR

(r=0.59 shown in Fig. 8c). A very similar west-east relationship also exists between the 15 EFR estimates

from GFAS and MDR (r=0.61, not shown). However, GFAS EFR values are generally lower across the

entire domain which can be traced back to differences in EFs (see Table 1). In general, we see lower MDR

and EFR values over the western part of the domain (where deforestation fires dominate) and higher MDR

and EFR values over the eastern part of the domain (where savanna fires are more prevalent). The

relationship becomes much more significant (r=0.89) if three outliers are excluded from the analysis. One

of these outliers represents a mountainous region in the southwest corner of the domain, where fire activity

was much lower than elsewhere in the domain. The other two outliers represent regions that are located

in between the tropical rain forest and the Cerrado savanna (highlighted in Fig. 8a and 8b). It is possible

that the MDR and EFR do not align well at these locations because the biome-specific EFs are not

representative for a more complex transition region. One would expect here more diversity in burning

practices, vegetation types and climate conditions, resulting in a mixture of different burning

characteristics that are not accounted for in the EF in GFED4s nor GFAS, which are based on coarser

resolution land cover data. Another factor that plays a role is atmospheric transport as it affects the column

mole densities that are measured downwind of dominant fire type. The wind direction was predominantly

from the east in September (see Fig. 4) and could as well carried the savanna-like combustion

characteristics from the most eastern regions more towards the west (see Fig. 8a).

### 3.2 Biomass burning characteristics across the world

We will discuss now more briefly the joint analysis of $\Delta XCO$ and $\Delta XNO_2$ for other regions around the world. This analysis is performed using SBM for multiple regions in Africa (Sect. 3.2.1), and using LSM for regions in central Australia (Sect. 3.2.2), Indonesia (Sect. 3.2.3) and boreal North America and Siberia (Sect. 3.2.4).

### 3.2.1 African savanna fires

The fire efficiency characteristics in the two regions in northern Africa (December 2018) and in the three regions in southern Africa (July-September 2018) are very similar to the fire characteristics in the South American Cerrado savanna discussed in detail in Sect. 3.1. In Fig. 9a we show $\Delta XCO$ and $\Delta XNO_2$ derived with SBM and the range in the monthly average MDR for the African regions. The day-to-day variations in $\Delta XCO$ and $\Delta XNO_2$ of all the months and regions combined form a well-defined cone-shaped envelope

that translates to MDR values ranging between 3.8 and 6.0. These values are higher than the average MDR derived for the Cerrado savanna in September (see Fig. 5c, MDR=2.8-3.2). Even though variations in fire intensities are substantial from month-to-month across the different savanna regions with different types of vegetation and climate conditions, we see in the TROPOMI data typically the same relatively small levels of $\Delta XCO$ and relatively high levels of $\Delta XNO_2$. This is again a strong indication that flaming

combustion is the most dominating fire phase in the savanna biome.

### 3.2.2 Australian savanna fires

The Australian savanna region consists mostly of arid regions with grasslands in the center of the continent (and where our focus lies) towards more densely vegetation woodlands in the north. Hot and



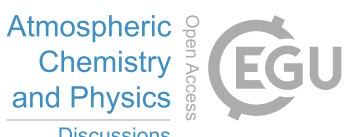

dry conditions in the southern hemisphere summer result in naturally occurring fires and prescribed burning as a fire management policy (Mallet et al., 2017). We show in Fig. 10a and 10b an example of XCO and $XNO_2$ for December 23, 2018, regridded at 0.1°×0.1° resolution. The plumes of XCO and $XNO_2$ that start at a number of hotspots and move in westward direction are easily recognizable. The background region for this particular day is located in the eastern part of the domain upwind of the

hotspots (shown by the purple box). The hotspots are positively identified as fires because their locations correspond very accurately with the locations of GFED4s and GFAS fire emissions (depicted by '+'-signs). One hotspot located in the most western part of the domain (west of 126°E) is not visible in TROPOMI XCO and $XNO_2$. The fire was likely short-lived and only detected in the morning with MODIS Terra satellite (local 10:30am overpass), three hours before the TROPOMI overpass. For the

other hotspots, Fig. 10a and 10b show a good correlation between the magnitude of the trace gas enhancements, the plume distances, and the spatial extent and magnitude of the fire emissions. In general, we observe higher trace gas enhancements for fire events detected over a larger surface area and for which we have larger emission estimates.

LSM was used for the Australian savanna region to derive ΔXCO and $ΔXNO_2$ in November and December 2018 (shown in Fig. 9b). The $ΔXNO_2$ levels range between 0 and 0.05 mmol $m^{-2}$ and the ΔXCO levels range between 0 and 0.01 mol $m^{-2}$. The average MDR over both months is 6.2 and therefore, as expected, the Australian savanna combustion resembles much more the efficient combustion characteristics measured for the South American and African savanna regions than the tropical

deforestation region (previously shown in Fig. 5).





### 3.2.3 Indonesian peatland fires

In Indonesia, peatland fires are a recurring seasonal phenomenon that generates severe atmospheric pollution and impacts public health (Marlier et al., 2013). In many lowland regions, forest clearing occurs along with drainage of peat-swamp forest exposing peat to fire. We investigated two peatland hotspots in

Indonesia using LSM: southern Borneo (Kalimantan) in September 2018 and western Sumatra in August 2018. In comparison with the savanna fires and deforestation fires discussed earlier, both peatland regions show a much larger contribution from smoldering combustion in the $\Delta XCO$ and $\Delta XNO_2$ data shown in Fig. 9c. The monthly average MDR derived over both regions is 1.0, which is approximately 4 and 1.5 times lower than the MDR derived for the savanna and tropical deforestation regions, respectively. These

results are consistent with the regional ground and airborne EF measurement campaigns. The peat itself burns almost entirely by smoldering combustion and generates a large amount of CO, which is evident in the large CO EF for peat (see Table 1). We expect our 2018 MDR estimate for Indonesia to be high in comparison to previous El Niño years like 1997-98 and 2015. In those years many fires spread out of control, consuming a significant portion of the surface vegetation and underlying peat soils causing a

severe deterioration in air quality (Page et al., 2002). The fraction of peat combustion in 2018 was likely smaller than during the El Niño years as it was generally colder and wetter in Indonesia. Therefore, the signal in MDR derived from TROPOMI accounts for a larger fraction of tropical forest overstory combustion, grassland, and agricultural fires which generate less CO than pure peat combustion.

### 3.2.4 Boreal forests

The observed combustion of boreal biomass (North America and Siberia) is considerably less efficient than in all other regions discussed. The relationship between $\Delta XCO$ and $\Delta XNO_2$ for these boreal regions,

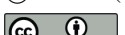

which is derived with LSM, is shown in Fig. 9d. During the dry season, the boreal forest fires consume

large amounts of above- and especially belowground biomass, including the burning of organic soils,

peat, and woody debris (Ottmar and Sandberg, 2003; French et al., 2004). Typically, such ecosystems

burn by residual smoldering combustion which can continue long after the initial flaming phase of the

fires (Akagi et al. 2011). It is apparent from the TROPOMI data that both boreal regions observed a 3 to

6 times larger increase in $\Delta$XCO relative to $\Delta$XNO$_2$ than the deforestation and savanna regions. It

consistently translates to a much lower monthly average MDR (<1) for the boreal fires on two different

continents, with higher MDR values over boreal North America compared to boreal Siberia.

**3.3 Global patterns in MDR and EFR**

By combining $\Delta$XCO and $\Delta$XNO$_2$ data from all investigated regions discussed in Sect. 3.1 and 3.2 we

can identify a four-way split in regional combustion activity and efficiency (see Fig. 11a). According to

the TROPOMI data, there is a group of boreal and peatland regions that emit relatively much CO per

gram of NO$_x$ in comparison to the other regions and are in the literature typically characterized as

smoldering fires. For these regions we determined MDR values ≤1.4. At the opposite end of the spectrum

are the savanna regions on three different continents that emit relatively much NO$_x$ generated from

nitrogen in the biomass and possibly from thermal decomposition of atmospheric N$_2$ at very high

combustion temperatures. The MDR values for savanna fires are much higher than for the boreal and

peatland fires, and range between 3.6-6.2 among the different savanna regions and sampling methods

(SBM for South America and Africa, LSM for Australia). In between these two extremes lie 4 different

South American deforestation regions with MDR values that range between 1.6 and 2.5, which is less

efficient than the savanna combustion but still more efficient than boreal and peat combustion.



A similar four-way split in regional combustion characteristics is shown in Fig. 11b using the four

synthetic WRF-CHEM simulations, each driven by a different set of modified biomass burning emissions

to mimic different ecosystems with frequent occurring fires (see Sect. 2.4). The estimates in ΔXCO and

ΔXNO₂ were derived with SBM and while day-to-day variability is quite substantial just like in the real

TROPOMI data in Fig. 11a, it demonstrates that the SBM method can provide a robust monthly average

estimate of MDR. The MDR estimates and the relative differences between the four different fire types

compare quite well with the actual derived signals in TROPOMI. The simulated savanna fires have

consistently the highest MDRs, which are about twice as high than the simulated tropical deforestation

fires, and 3 to 4 times as high than the simulated peatland and boreal fires, respectively.

The pattern of combustion signatures in MDR determined with TROPOMI across the different fire types

compare well with the spatial patterns in GFED4s EFR (see Fig. 12a and b). The Pearson correlation

coefficient between estimates of MDR and EFR across the 5 continental areas implies a strong linear

relationship exist (r=0.90). It demonstrates that the regional combustion efficiency that is detected with

TROPOMI is generally consistent with the worldwide spatial distribution of EFs used by GFED4s (mostly

based on Akagi et al., 2011). For the savanna fires we have consistently the highest estimates for MDR

and EFR. The MDR and EFR for tropical deforestation fires are about half those of savanna fires. The

peatland fires in turn have 3 times lower MDR and EFR. The lowest values in MDR and EFR (4 times

lower than savanna fires), which suggest the least efficient type of biomass combustion, is shown for the

North American and Siberian boreal regions. Note that the retrieved linear relationship in Fig. 12b

between GFED4s EFR and TROPOMI MDR is similar to the linear relationship and slope found between

EFR and MDR based on the 4 synthetic WRF-CHEM simulations. Even with the aforementioned caveat that these simulations are simple in design, it does demonstrate quite convincingly that satellite and ground-based measurements of trace gas ratios are related to one another through a simple linear relationship.

The pattern of combustion signatures in MDR is somewhat different from EFR of GFAS (see Fig. 12a and c), which is mostly based on the older set EFs from Andreae and Merlet (2001). The Pearson correlation coefficient between MDR and EFR is lower (r=0.49) and the slope across the different fire regions is less steep. One reason is that the boreal fire characteristics in GFAS were lumped together with the temperate fires into a single category called extratropical fires, which reflects a much smaller

smoldering combustion component and thus a higher EFR (see Table 1). As a consequence, Siberian boreal $NO_x$ emissions are in GFAS approximately four times larger than in GFED4s. This confirms the findings in the CAMS validation activity: Comparisons with GOME-2 indicated largely overestimated Boreal $NO_2$ concentrations in the CAMS forecasts driven by GFAS (Ramonet et al., 2019). Another reason for the mismatch is that the combustion efficiency of savanna fires in GFAS is of the same order

of magnitude as the combustion efficiency in the boreal regions, which seems less realistic given the current body of EFs measurements from savanna ecosystems that claim the contrary. For peatland fires in Borneo, Indonesia, GFAS assumes a much larger fraction of smoldering combustion than GFED4s also. This large fraction of smoldering combustion would probably be more accurate during the drier years, e.g. during El Niño, than for the relatively wet year 2018.

## 4 Discussion

In this paper, we demonstrated the capability of new high-quality XCO and XNO$_2$ column observations from the space-borne TROPOMI instrument to detect and quantify spatial variations in biomass combustion efficiency from a top-down perspective. The TROPOMI observations of XCO and XNO$_2$ (a good proxy for NO$_x$) and the Mole Density Ratio (MDR) between the local enhancements of the two species have an important advantage over ground or airborne-based measurements due to the daily global spatial coverage, and as such are complementary to bottom-up derived EFR signatures.

We found distinct spatial patterns in MDR across different regions and continents which signify very different combustion efficiency characteristics. Irrespective of the utilized sampling method, these patterns in MDR compare well with EFR signatures around the world from existing fire emission datasets. In principle, these findings are not new but confirm from a remote sensing perspective the general spatial distribution of combustion efficiency of the current body of EF measurements. Based on the TROPOMI measurements of just one year we derived a first (but still preliminary) linear relationship between TROPOMI column observations of CO and NO$_2$, and EFs of CO and NO$_x$ near the fire source (see Fig. 12b). This approach provides an additional anchor to help constrain combustion characteristics and can for instance be used to estimate and quantify the spatiotemporal variability of combustion efficiency around the world, even over regions where there is a deficiency of detailed information on fuel load, combustion practice and EFs.



The combustion in all savanna regions in South America, Africa and Australia were consistently cleaner and more efficient (i.e. highest MDR) than for all the other regions investigated. The two different methods of sampling provided quite similar MDR estimates. The MDR for Australia was determined by studying individual fire plumes with LSM and yielded the highest MDR estimate, but also the largest day-to-day uncertainty. The other three savanna regions were determined with SBM and yielded smaller

MDR values but were still significantly larger than the MDRs derived for the other fire types. The LSM method may be inclined to higher MDRs because $\Delta XNO_2$ is derived from mole density measurements in close proximity of the actual fire sources where $XNO_2$ is at its highest level before any significant removal with OH occurs. On the other hand, the area sampled was very arid and mostly consisted of grasses. Therefore, smoldering combustion products like CO and $CH_4$ tend to be lower (Hurst et al., 1994). Our

study also shows a relatively clean combustion process for the Northern and Southern African savanna fires, in agreement with the current EF datasets. In contrast to Zheng et al. (2018), we did not find evidence of a seasonal transition from flaming to smoldering combustion for the different African regions. They inverted multi-year XCO column measurements from the MOPITT instrument and found that GFED4s significantly underestimates the CO emissions by 12 to 62% later in the fire season. They partly attributed

this outcome to the static EFs that are currently in use that omit seasonal variations in burning conditions. We therefore argue that the underestimation of GFED4s CO emissions is more likely the result of missing burned area detections in the late dry season.

In comparison to the savanna fires, lower MDR values were derived for the South American deforestation

regions, indicating a larger contribution from smoldering combustion of organic soils and woody debris

that is piled together at the surface. These spatial differences in combustion efficiency between

deforestation and savanna fires agree with the study of Silva and Arellano (2017), however, a one-on-one

comparison between the two studies is difficult. They derived estimates of MDR based on the ratio of

$\Delta XCO/\Delta XNO_2$ (instead of $\Delta XNO_2/\Delta XCO$), and they did their analysis for a different year, probably

under somewhat different meteorological and chemistry regimes.

The least efficient type of combustion with the lowest MDR values were detected for the Indonesian

peatland fires and boreal fires of North America and Siberia. These fires characteristically spread as

smoldering fires by burning peat, organic soils and woody debris at the surface. The difference in MDR

between the North American and Siberian boreal fires in Fig. 12a suggest different fire dynamics between

the two boreal regions. These differences do not appear in the EFR estimates of GFAS and GFED4s

because the EF datasets lack spatial and temporal variability for each fire type. The lower average MDR

value for Siberia indicates generally more smoldering combustion (less $NO_x$, more CO) than the

combustion in North America. This result supports independently the findings of Wooster and Zhang

(2004) and Rogers et al. (2015), who found compelling evidence for smaller fire intensity and burn

severity in the Siberian boreal forests across multiple satellite datasets (but not TROPOMI) and forest

inventories. Rogers et al. (2015) related the differences in fire dynamics between these two regions to

their dominant tree species. Pine trees in Eurasia have evolved to resist and suppress crown fires.

Therefore, the fires in these areas are usually reported as surface fires, which burn mostly in the

smoldering phase. The trees in the northern parts of North America have evolved to spread and be

consumed by more intense crown fires, killing most trees. Yet, we remain cautious to fully attribute the

detected differences between the North American and Siberian MDR to the burning characteristics of specific tree species until we have analyzed multiple years of TROPOMI data. The uncertainties in MDR for 2018 (based on day-to-day variability) are still quite substantial for the boreal regions as is shown in

Fig. 11 and Fig. 12a.

Our estimates of MDR across the world compared most favorably with the spatial distribution of biomass burning efficiency prescribed in GFED4s, where the proxy of efficiency is carried through the EFs of CO and $NO_x$. The remotely sensed measurements confirm the addition of a dedicated boreal forest fire type

as a key improvement that was implemented in the more recent Akagi et al. (2011) EF database (used by GFED4s). It underlines the need for EFs that reflect a large component of smoldering combustion of organic soils and boreal peat in this part of the world (Yokelson et al., 1997; Bertschi et al., 2003). This was specifically done in the Akagi et al. (2011) EF dataset where they applied an equal weighting scheme for the boreal region airborne measurements (which have a bias towards flaming fires) and ground-based

measurements (which have a bias towards smoldering fires). Our MDR estimate for the two Indonesian peatland regions is lower than MDR for deforestation fires but higher than the MDR for boreal fires. Therefore, it is likely that it represents a combination of peat soil combustion (usually consumed almost entirely by smoldering) and overstory combustion of tropical forests, grasslands, and agriculture (usually consumed by flaming and smoldering). This relative pattern is in agreement with a more mixed

combustion efficiency prescribed in GFED4s for Indonesia (both for Borneo and Sumatra). The Borneo fires in GFAS exhibited a much larger smoldering component, which may be more common during El Niño years when fires spread out of control, consuming a significant portion of the underlying peat soils.

However, 2018 was not an El Niño year, and evidence of excessive smoldering combustion was not found in the TROPOMI data (see Fig. 11, 12a and c). In fact, the TROPOMI data suggest more complete
combustion efficiency in Borneo than in Sumatra (1.43 vs. 0.94 MDR).

The day-to-day variations in MDR (shown in Fig. 11 and depicted by the 1σ error bars in Fig. 12a) point to a considerable amount of uncertainty. An important source of this uncertainty is first of all the SBM sampling method. Estimates of $\Delta$XCO and $\Delta$XNO$_2$ can deviate substantially on a daily basis depending
on how much the sampled data is skewed to either side of the scale, away from a perfect Gaussian normal distribution. It was demonstrated for South American fires (see Fig. 5) that the SBM and the two alternative sampling methods can produce quite a range in MDR. Similarly, the instrument/retrieval precision of TROPOMI's XNO$_2$ was also a small source for uncertainty in MDR (see Fig. 6). However, regardless of the sampling method or precision, we were still able to distinguish clearly the deforestation
fires from savanna fires using the monthly aggregated $\Delta$XCO and $\Delta$XNO$_2$ data. The alternative sampling methods were also used to derive MDR from the synthetic WRF-CHEM simulations, and similarly, it did not deteriorate our ability to distinguish the four different fire types (not shown in paper).

Another source of uncertainty in MDR is the difference in surface sensitivity of TROPOMI's XCO and
XNO$_2$ measurements. A comparison of the column averaging kernel (AK) of both species shows that tropospheric XNO$_2$ measurements are generally less sensitive to sources in the planetary boundary layer than XCO measurements. From the surface to approximately 800hPa is the sensitivity of XNO$_2$ smaller than for XCO but increases from the mid-troposphere to tropopause (800-200hPa).  This is one of the



reasons why our daily estimates of ΔXNO₂ are biased low. Potentially, it has an effect on most of our

MDR estimates because it has been demonstrated, using stereo-height measurements of smoke plumes, that most fires are typically emitted inside the planetary boundary layer (Martin et al., 2018). The estimates of ΔXCO, ΔXNO₂ and MDR derived from the simulated XCO and XNO₂ column data (WRF-CHEM experiments in Fig. 11 and 12) were calculated with a daily region-average AK for September 2018. This provided more realistic column estimates for both species (and thus a more realistic MDR

estimate) that allowed a better one-to-one comparison to TROPOMI MDR estimates, even under cloudy conditions (Borsdorff et al. 2018b). Not using the AKs to derive MDR with WRF-CHEM, and instead simply using the simulated total XCO and the tropospheric XNO₂ column densities, would yield higher MDR estimate. This is because simulated XNO₂ enhancements from surface fire sources are, in comparison to TROPOMI's limited measurement sensitivity in the PBL, unrealistically overrepresented

in WRF-CHEM.

In general, a large part of the biases in ΔXNO₂ (and thus in MDR), either caused by the sampling techniques or the precision and sensitivity, were in all likelihood somewhat similar in magnitude in the regions we studied. Hence, we believe it did not impair the detection of differences in fire characteristics.

The uncertainty related to chemistry and transport may have played a larger role region-to-region as it affected tropospheric NO₂ more differently than CO, and thus our ability to derive a robust MDR. In particular, on shorter day-to-day time scales the MDR estimates can vary greatly. The amount of OH radicals in the atmosphere acts as the primary daytime sink of NO₂ and can vary substantially depending on the amount of tropospheric O₃, water vapor and incoming sunlight (source of OH), and the presence

of other chemical species such as volatile organic compounds (sink of OH). Overall, it reduces the lifetime

of $NO_2$ to several hours, much shorter than the lifetime of CO. As a consequence, daily estimates of

$\Delta XNO_2$ will always be biased low. In addition, daily variations in $\Delta XNO_2$ that are driven by transport

and chemistry are naturally exacerbated in $\Delta XNO_2/\Delta XCO$ ratio-space. Therefore, to interpret MDR, it is

currently necessary to collect multiple days of data (e.g. for an entire month) to retrieve a more robust

combustion efficiency signature that cancels out some of the day-to-day variations in transport and

chemistry. Potentially we could minimize these variations retroactively by inverting the measured MDR

back to a daily EFR estimate, where we take the removal of $NO_2$ into account. This could provide a more

direct top down estimate of EFR and could improve the detection of seasonal (and maybe even daily)

changes in fire characteristics. For instance, the transition from flaming to the more smoldering fires, as

suggested to occur in the African savanna (Zheng et al., 2018) or the supposed differences between North

American and Siberian boreal fires (Rogers et al., 2015) might be detected more easily that way. Future

research could explore this but requires a more elaborate analysis for each region separately, with

emphasis on acquiring a better understanding of the daily variations of regional OH content, windspeed

and direction, and the chemical rate constant of $NO_2$ removal.

**5 Conclusion**

We have investigated regional biomass burning characteristics and efficiency using the new space-based

TROPOMI measurements of XCO and $XNO_2$. The mole density ratios (MDR) between regional

enhancements of $XNO_2$ and XCO have been quantified using different sampling techniques, which have

been tested using WRF-CHEM simulations accounting for realistic atmospheric transport, chemistry and

the limited instrument sensitivity to the lower atmosphere. TROPOMI provides independent support for

the more recent Akagi et al. (2011) set of EFs used in fire emission products like GFED4s. We have found

spatial variations in combustion efficiency that match the ground and airborne measurements of EFs quite

accurately. Generally, boreal fires show a much larger fraction of smoldering combustion than savanna

grassland and tropical deforestation fires (boreal ecosystem cause a 3 to 6 larger increase in $\Delta$XCO than

$\Delta$XNO$_2$). On smaller spatial scales of a thousand kilometers, we also found gradients in combustion

characteristics from west to east over Brazil. In the state of Amazonas, the practice of tropical

deforestation, where woody debris after initial clearing is ignited during the dry season, is clearly

distinguishable from fires in the savanna-like ecosystem in central Brazil, where fires mostly consume

the grass-layer by flaming combustion. We have found deforestation fires to cause a 1.5 to 2 times larger

increase in $\Delta$XCO relatively to $\Delta$XNO$_2$ than the savanna fires, mainly because these fires reflect a larger

fraction of surface smoldering combustion. The detected differences interregional (e.g. boreal vs.

savanna) and intraregional (e.g. North America vs. Eurasia boreal region) underline that TROPOMI can

provide new top-down constraints on biomass burning characteristics and EFs.



## Data availability

TROPOMI measurements of $NO_2$ and CO can be downloaded from https://s5phub.copernicus.eu; GFED4s fire emissions can be downloaded from https://www.geo.vu.nl/~gwerf/GFED/GFED4/; GFAS fire emissions can be downloaded from https://apps.ecmwf.int/datasets/data/cams-gfas/.

## Code availability

WRF-CHEM atmospheric transport model version 4.0 can be downloaded from https://www2.mmm.ucar.edu/wrf/users/downloads.html.

## Author Contributions

IvdV did the data analysis, designed and ran the model simulations and wrote the paper. GvdW, SH, and IA provided science advise and detailed comments on the manuscript. HE, JPV, and TB provided additional comments on the manuscript and TROPOMI retrieval products.

## Competing interests

The authors declare that they have no conflict of interest.

## Acknowledgements

We would like to thank the team that realized the TROPOMI instrument, consisting of the partnership

between Airbus Defence and Space Netherlands, Royal Netherlands Meteorological Institute KNMI,



SRON Netherlands Institute for Space Research, and the Netherlands organisation for applied scientific research (TNO), commissioned by the Netherlands Space Office (NSO) and European Space Agency (ESA). Sentinel-5 Precursor is part of the EU Copernicus program, and Copernicus Sentinel data 2018 has been used. The WRF model computations were carried out on the Dutch national supercomputer

Cartesius maintained by SURFSara (www.surfsara.nl). GvdW has been supported by Netherlands organization for Scientific Research (NWO, VICI research program 016.160.324).





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



**Table 1: Emission factors for NO$_x$ (emitted as NO) and CO used by GFED4s (mostly based on Akagi et al., 2011 and a few other sources) and GFAS (mostly based Andreae and Merlet. 2001 with additional updates) for different types of biomass burning. The final two columns on the right show the ratio between EF$_{NOx}$ and EF$_{CO}$ (EFR) for the two emission databases. The original units [g kg$^{-1}$] are converted to [mmol kg$^{-1}$] and [mol kg$^{-1}$] for EF$_{NOx}$ and EF$_{CO}$, respectively, to make units of EFR equal to TROPOMI mole density ratios.**

| | EF$_{NOx}$ [mmol kg$^{-1}$] | | EF$_{CO}$ [mol kg$^{-1}$] | | EFR = EF$_{NOx}$/EF$_{CO}$ | |
|---|---|---|---|---|---|---|
| | GFED4s | GFAS | GFED4s | GFAS | GFED4s | GFAS |
| Peat fires | 33.33 | 33.33 | 7.50 | 7.50 | 4.44 | 4.44 |
| Boreal forest fires | 30.00 | - | 4.54 | - | 6.61 | - |
| Temperate forest fires | 64.00 | - | 3.14 | - | 20.38 | - |
| Extratropical fires | - | 113.33 | - | 3.79 | - | 29.90 |
| Tropical deforestation fires | 85.00 | 82.14 | 3.32 | 3.61 | 25.60 | 22.75 |
| Agricultural waste burning | 103.67 | 82.14 | 3.64 | 3.29 | 28.48 | 24.97 |
| Savanna fires | 130.00 | 70.00 | 2.25 | 2.18 | 57.78 | 32.11 |

080

085



**(a) GFED4s Emission Factor Ratio**

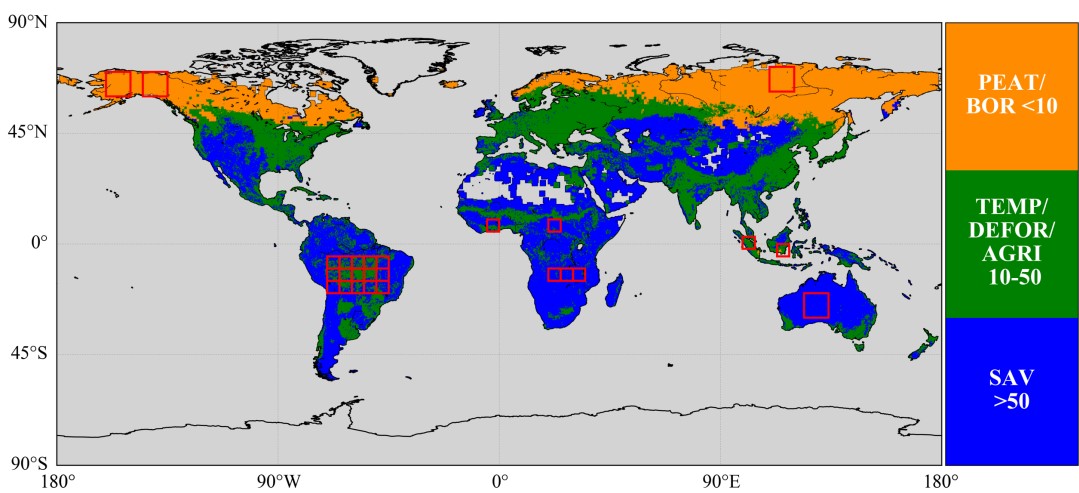

**(b) GFAS Emission Factor Ratio**

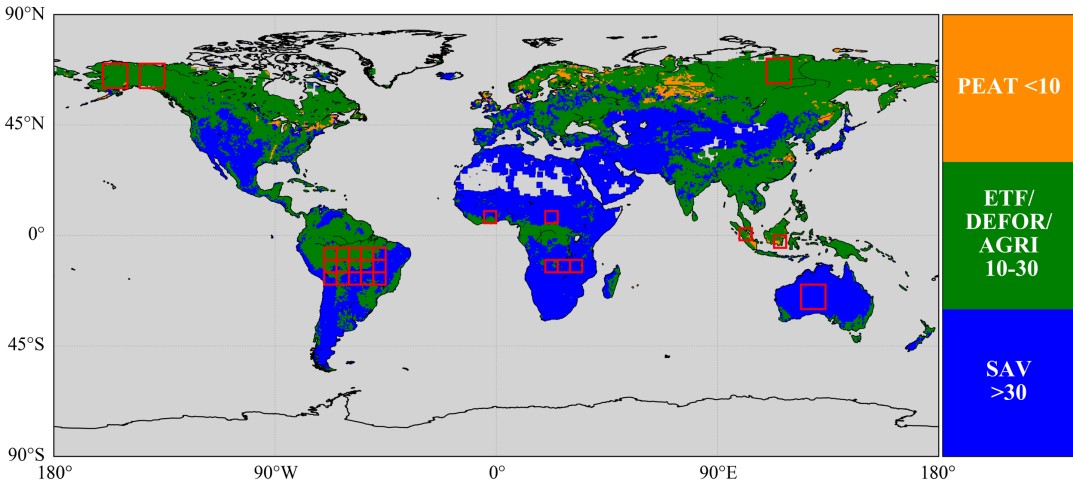

**Figure 1: Emission factor ratio between NOx and CO (EFR) for GFED4s (a) and GFAS (b). In panel (a) the range of EFR values are subdivided into three aggregated fire type categories: (1) peat and boreal fires (PEAT/BOR: <10), (2) temperate forest, deforestation and agricultural fires (TEMP/DEFOR/AGRI: between 10 and 50) and (3) savanna fires (SAV: >50). In panel (b) the range of EFR values subdivided along a different classification (see Sect. 2.1): (1) peat fires (PEAT: <10), (2) extratropical forest, tropical deforestation and agricultural fires (ETF/DEFOR/AGRI: between 10 and 30), and (3) savanna fires (SAV: >30). Regions of interest are highlighted by the red boxes.**





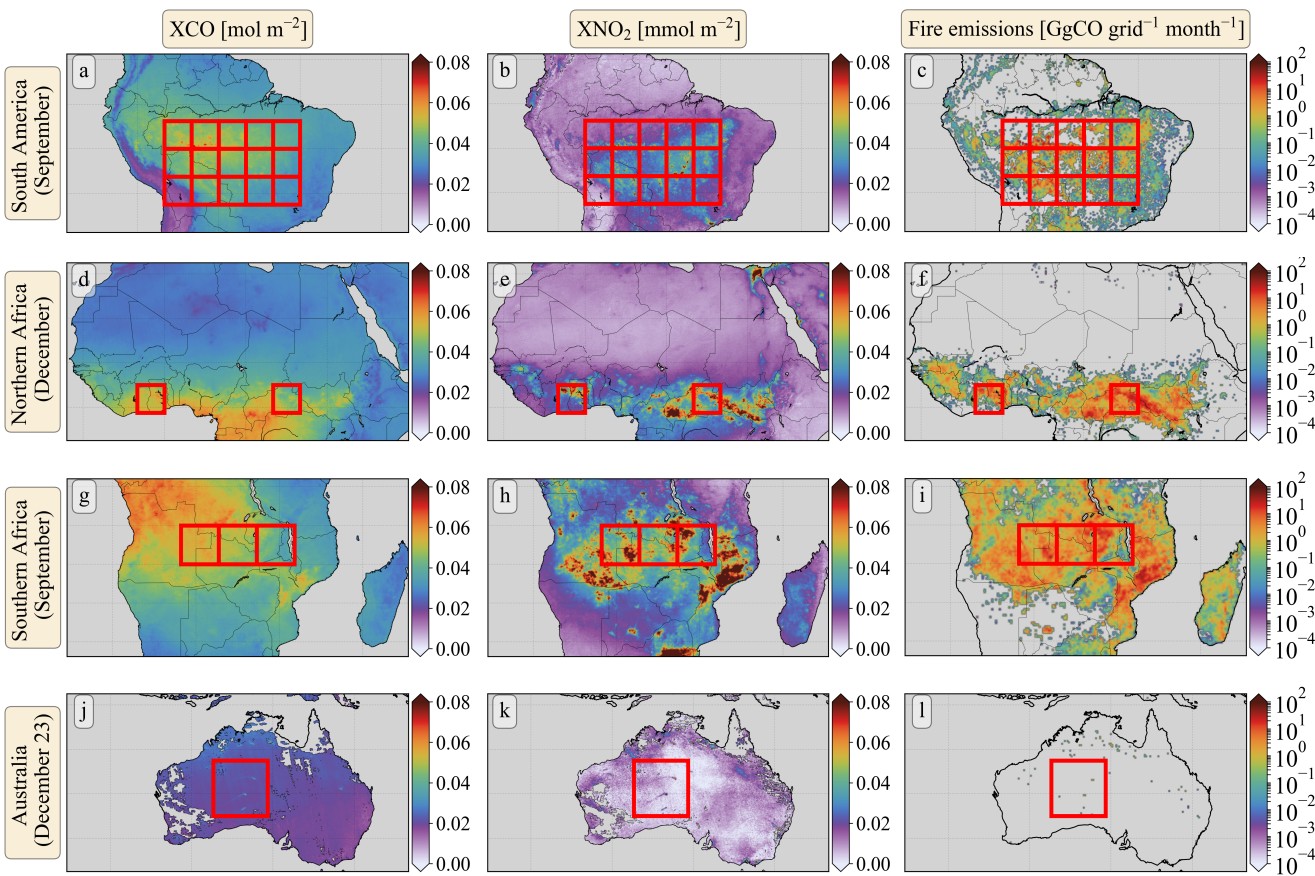

**Figure 2:** Maps of monthly average XCO [mol m$^{-2}$], XNO$_2$ [mmol m$^{-2}$], and GFED4s CO emissions [GgCO grid$^{-1}$ month$^{-1}$] for South America (panels a-c), for Northern Africa (panels d-f), and for Southern Africa (panels g-i). Maps of daily average XCO, XNO$_2$ and GFED4s CO emissions are shown for Australia for December 23, 2018 (panels j-l). Regions of interest are highlighted by the red boxes.





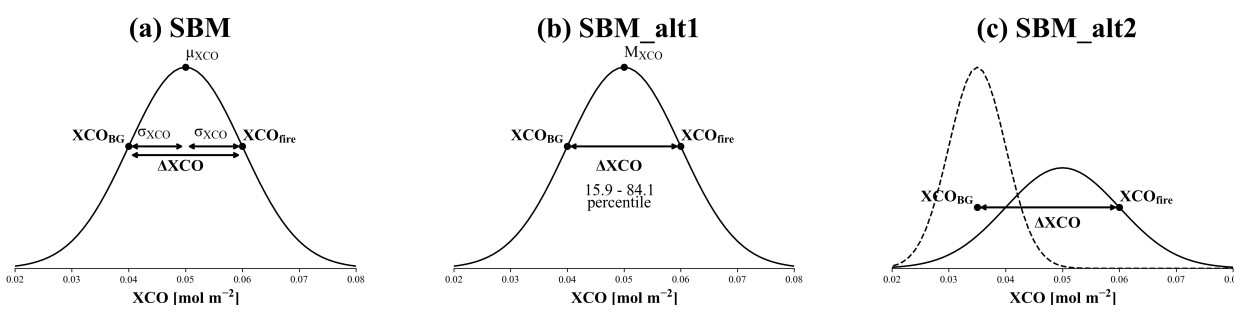

**Figure 3: Three types of the statistical bulk method (SBM) are applied to an idealized distribution of XCO samples [mol m$^{-2}$]. In panel (a) the standard SBM where ΔXCO (XCO$_{fire}$ − XCO$_{BG}$) is equal to 2σ$_{XCO}$ around the distribution mean μ$_{XCO}$. In panel (b) the first alternative SBM (SBM_alt1) where ΔXCO is equal to the difference between the 84.1 percentile rank and the 15.9 percentile rank around the median M$_{XCO}$. The estimates for ΔXCO from SBM and SBM_alt1 are only equal if the distributions are perfectly Gaussian. In panel (c) the second alternative SBM (SBM_alt2) where ΔXCO is equal to the difference between XCO$_{fire}$ from the standard SBM and XCO$_{BG}$ derived from the mean concentration of another XCO distribution sampled upwind of the fire region (dashed distribution). The estimates for ΔXCO from SBM and SBM_alt2 are only equal if XCO$_{BG}$ are the same.**





**Sample regions, WRF-CHEM domain, and PBL wind field**

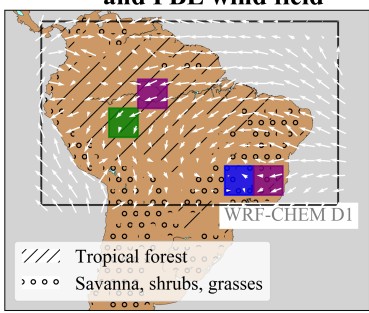


**Figure 4: Location of the deforestation 5°×5° sampling region (green), the savanna 5°×5° sampling region (blue) and two background regions adjacent to the two source regions (purple). The WRF-CHEM domain and the predominant wind direction in the PBL during the 2018 fire season are superimposed. The location of the green region is also used to sample XCO and XNO₂ data from four WRF-CHEM simulations (see Sect 2.4).**







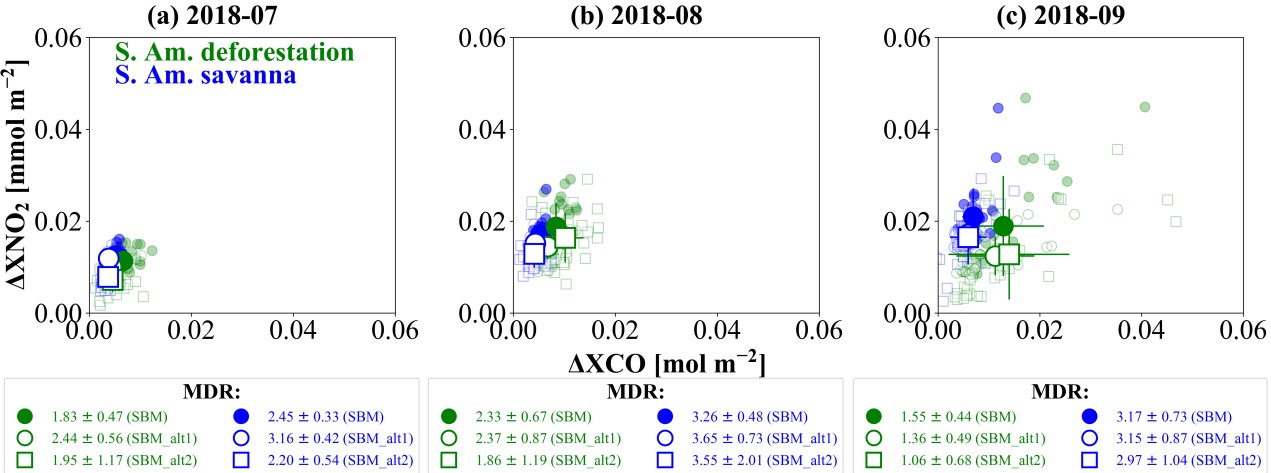

Figure 5: The relationship between daily ΔXCO [mol m⁻²] and ΔXNO₂ [mmol m⁻²] from TROPOMI for the South American deforestation (green) and savanna (blue) region for three consecutive months in 2018: July in panel (a), August in panel (b) and September in panel (c). For each month estimates are shown for three different sampling methods: SBM (solid circles), SBM_alt1 (open circles), and SBM_alt2 (open squares). In addition, the monthly average relationship between ΔXCO and ΔXNO₂ is depicted by the bigger markers together with error bars to indicate the 1σ day-to-day variability. The legend of each panel includes monthly average MDR and 1σ day-to-day variability estimates for each region and sampling method.





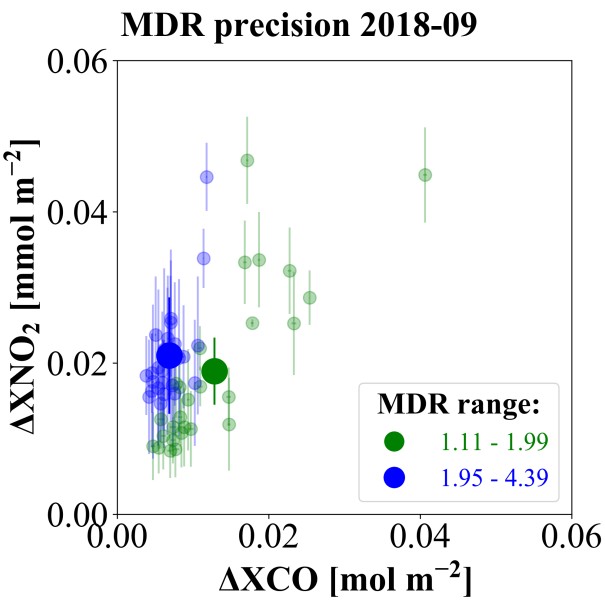

**Figure 6:** The relationship between daily $\Delta$XCO [mol m$^{-2}$] and $\Delta$XNO$_2$ [mmol m$^{-2}$] from TROPOMI for the South American deforestation (green) and savanna (blue) region for September 2018 using SBM. The error bars depict the range in $\Delta$XCO and $\Delta$XNO$_2$ estimates induced by the instrument measurement precision. The range in $\Delta$XNO$_2$ is larger than $\Delta$XCO because relatively, the XNO$_2$ instrument precision is less accurate than XCO. The legend reports the range of values in the monthly average MDR for both regions.







**Figure 7: September 2018 daily estimates of ΔXCO [mol m⁻²] and ΔXNO₂ [mmol m⁻²] from TROPOMI derived with SBM, together with smoothed estimates of the MDR (colored dashed line), and smoothed CO emission estimates [GgCO region⁻¹ day⁻¹] from GFED4s (black solid line) and GFAS (black dashed line). Panel (a) shows the results for the South American deforestation region and panel (b) for the South American savanna region.**





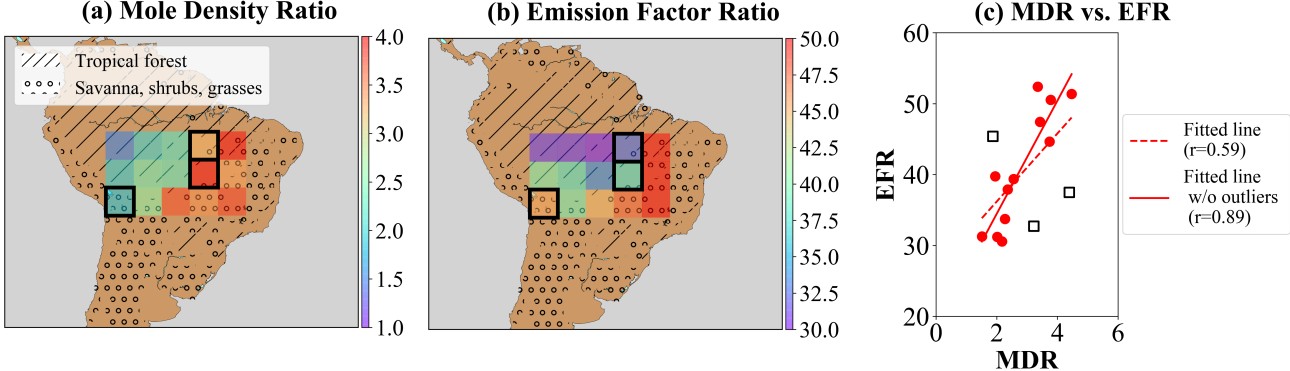


**Figure 8: In panel (a) the spatial pattern of the September average MDR determined with TROPOMI for 15 5°×5° regions using SBM. In panel (b) for the same 15 regions the spatial pattern of the September average EFR determined from the ratio between NO$_x$ and CO emissions from GFED4s. Panel (c) shows the relationship between the 15 MDR and EFR estimates, including the dashed linear regression line and the Pearson correlation coefficient. The three largest outliers are identified with a black square symbol and are highlighted by the black frames in panels (a) and (b). The linear regression line without the outliers is shown by the solid line in panel (c).**




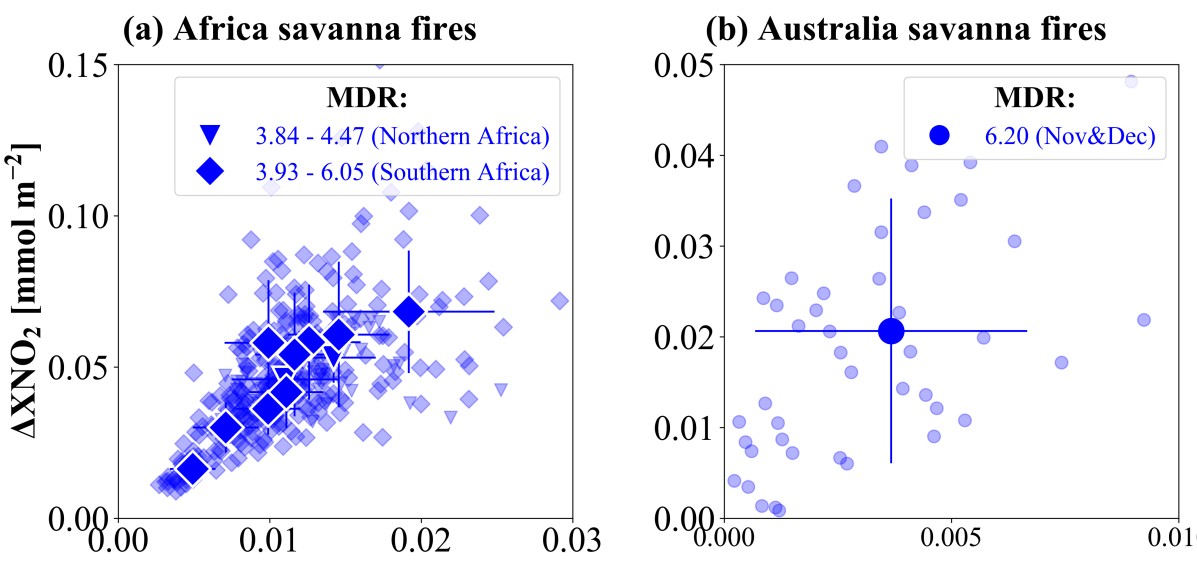


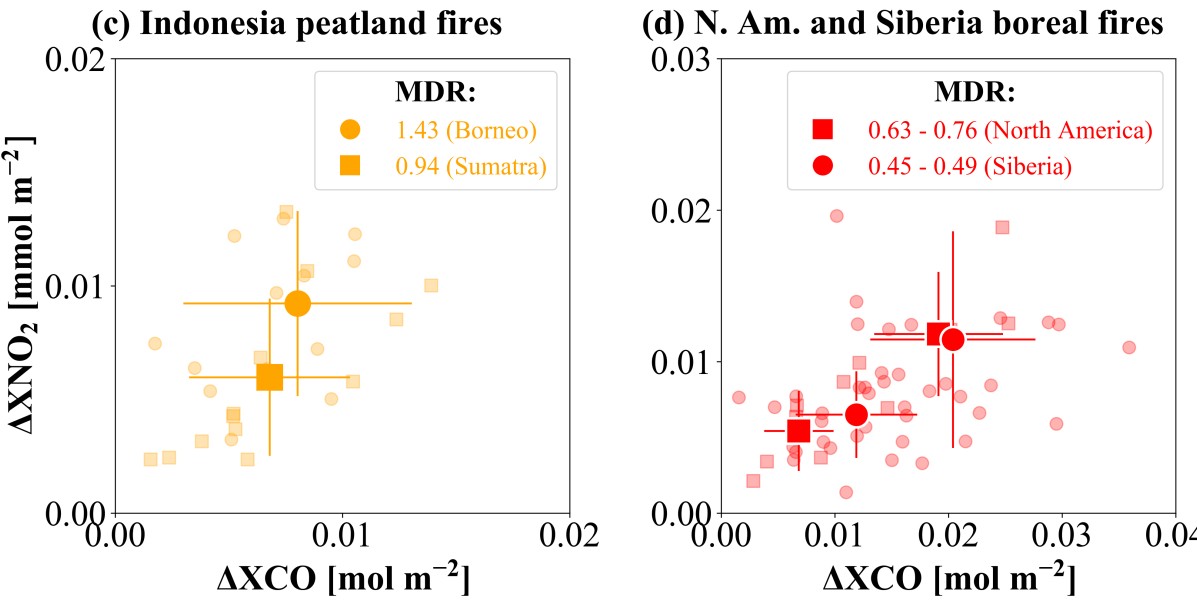

**Figure 9: The relationship between daily ΔXCO [mol m⁻²] and ΔXNO2 [mmol m⁻²] from TROPOMI derived for the different regions around the globe. In panel (a) the northern (red) and southern (blue) African savanna regions for different months during the 2018 fire seasons. In panel (b) the central Australia savanna region for November and December 2018. In panel (c) the Indonesian Borneo peatland region for September 2018 and Sumatra peatland region for August 2018. In panel (d) the two North American boreal regions for July 2018 and Siberian boreal region for July and August 2018. The monthly average relationship between ΔXCO and ΔXNO2 for the different regions and months is depicted by the big markers together with error bars to indicate the 1σ day-to-day variability. The legend includes the monthly average MDR estimates for the different regions.**



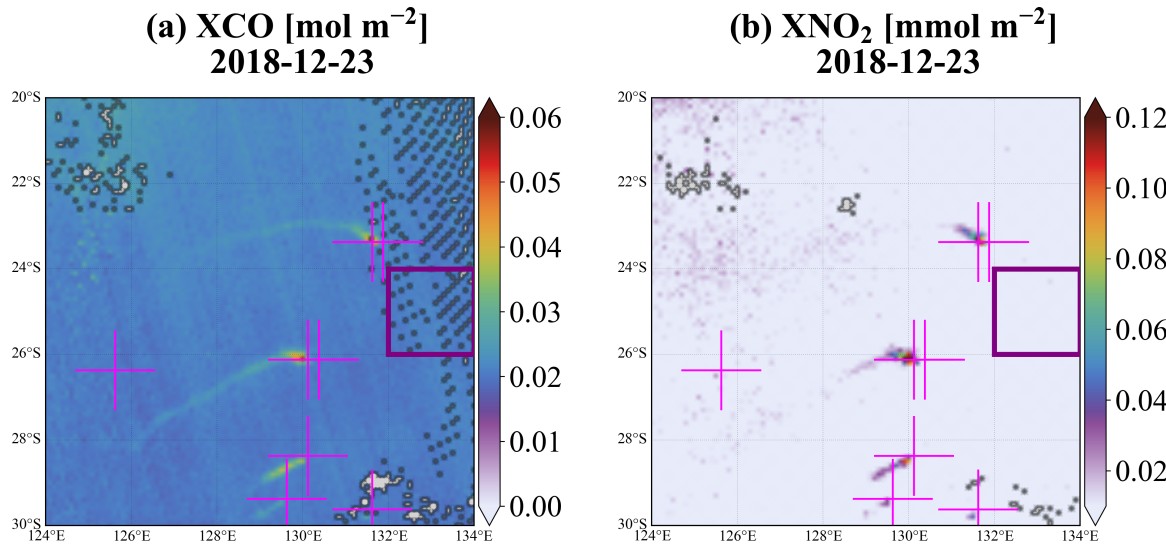


**Figure 10:** The column density XCO [mol m⁻²] in panel (a) and XNO₂ [mmol m⁻²] in panel (b) measured with TROPOMI on December 23, 2018 over central Australia. The background region is depicted by the purple frame and is located upwind of multiple fire plumes. The locations of the GFED4s/GFAS fire emissions are depicted by the magenta plus-signs.





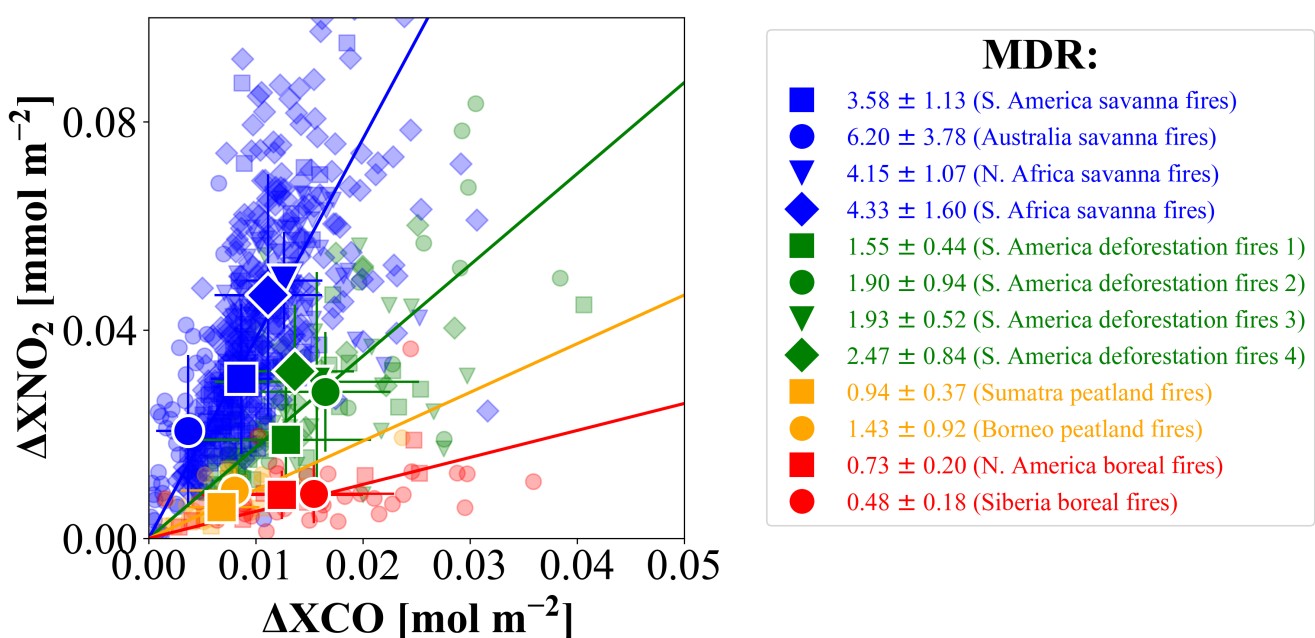

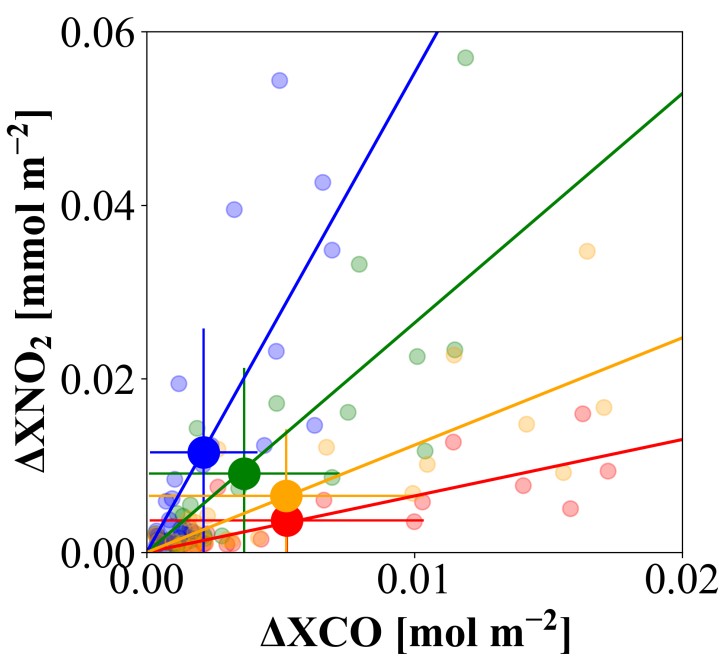

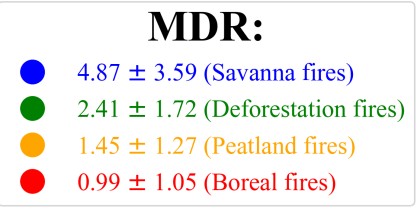

**Figure 11: In panel (a), the relationship between daily $\Delta$XCO [mol m$^{-2}$] and $\Delta$XNO$_2$ [mmol m$^{-2}$] from TROPOMI combined for all regions of this study. The region average relationship between $\Delta$XCO and $\Delta$XNO$_2$ is depicted by the big markers together with error bars to indicate the 1$\sigma$ day-to-day variability. The four regression slopes with intercept at zero signifies the different groupings of combustion efficiency. The legend includes the average MDR and 1$\sigma$ day-to-day variability estimates for each region. For the South American deforestation regions, we used data in September, at the height of the 2018 fire season. In panel (b), the relationship between daily $\Delta$XCO and $\Delta$XNO$_2$, and the monthly average MDR from four different synthetic WRF-CHEM simulations (see Sect. 2.4).**




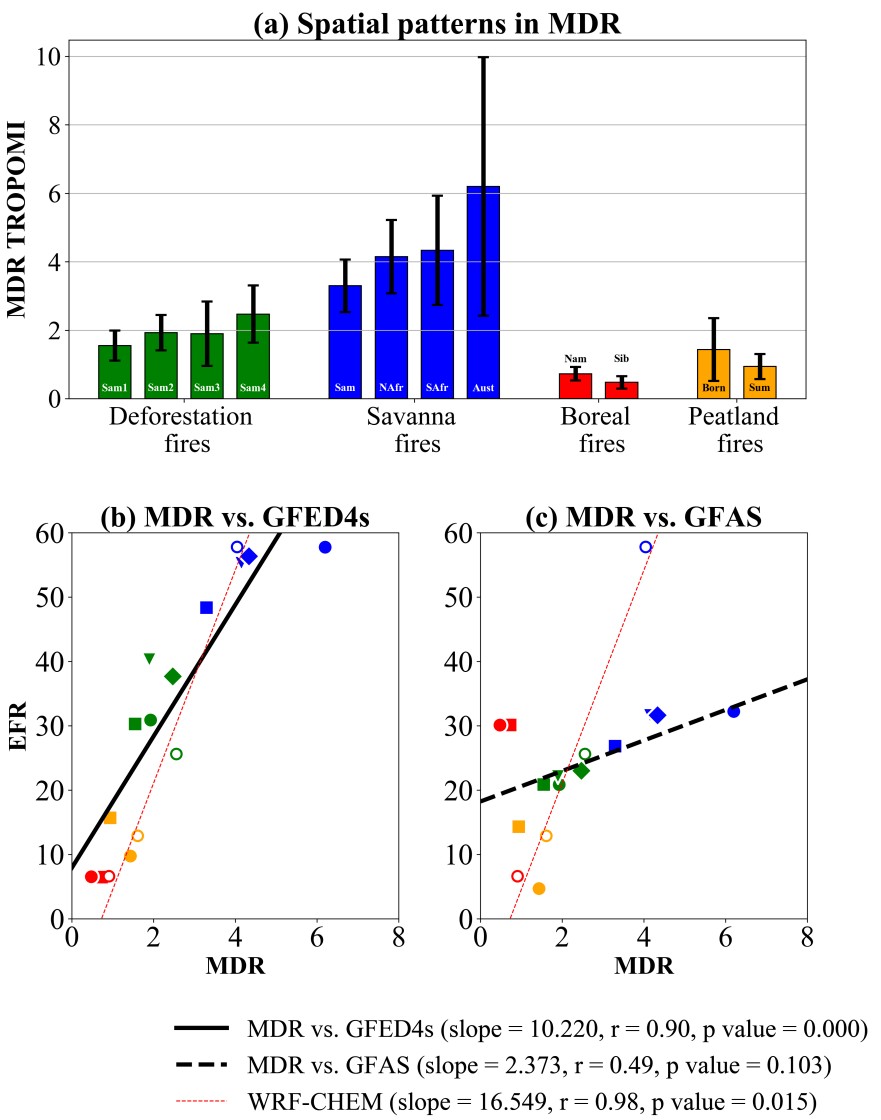

MDR vs. GFED4s (slope = 10.220, r = 0.90, p value = 0.000)

MDR vs. GFAS (slope = 2.373, r = 0.49, p value = 0.103)

WRF-CHEM (slope = 16.549, r = 0.98, p value = 0.015)

**Figure 12:** In panel (a) the inter- and intraregional comparison of MDR (colored bars) with error bars depicting 1σ day-to-day variability. The relationship between the average MDR and EFR from GFED4s is shown in panel (b), and between the average MDR and EFR from GFAS in panel (c). The different colored symbols correspond with regions listed in the legend of Fig. 11a. The linear regression derived from the MDR-EFR data is shown for both panels (black solid and dashed lines). The MDR-EFR relationship from the four WRF-CHEM simulations is shown by the colored open circles and the linear regression through these four markers is shown by the red dashed line. The slope, Pearson correlation coefficient and the two-sided p-value (for a hypothesis test whose null hypothesis is that the slope is zero) is reported for each regression line.