# Peer review of "Biomass burning combustion efficiency observed from space using measurements of CO and NO2 by TROPOMI"

_Atmospheric Chemistry and Physics, 2020_

## Referee Comment (RC1) · Anonymous Referee #1

2020-05-19
10.5194/acp-2020-272-RC1
Author(s) 2020

---

## Referee Comment (RC2) · Anonymous Referee #2 · 7 Jun 2020

This work has carefully investigated to obtain emission factors for biomass burning combustion from the satellite instruments with a large amount of information provided. I think this work contains useful information on biomass burning, which can specify many types of burning globally. Despite many information provided and careful investigation, it needs to be shortened to focus on the main findings. The lengthy descriptions of each session can distract the main points of the originality of the work.

Specific comments

The ratio between XNO2 and XCO implies not only the information on the surface emissions but also the information of its transport especially considering the longer

lifetime of CO. How do you think the column comparison can cause the uncertainties of surface emissions? The author may have to comment on this concisely.

Are the types of burning affected by the soil and its condition? I wonder if more factors impact on the burning conditions (XNO2 and XCO ratios).

I think the information in Figure 11 and 12 is better in the table. The table of EFs would be useful with the regions, types of burning, and seasons. That would be useful for scientific communities.

Are the ratios of deforestation fires different from all types of vegetation fires? Then it means we can capture the deforestation by MDR from space? Please describe the meaning of identifying the deforestation by the satellite sensing.
* * *

---

## Author Response (AR1)

Reviewer #1

We kindly thank the reviewer for his/her time to evaluate our manuscript, and we appreciate the positive view on the work presented. Our comments appear below in red.

Specific comments:

The authors designed four synthetic WRF-Chem experiments driven by different biomass burning scenarios representative of different regions but all the simulations were performed over South America. Since the chemistry and weather of boreal forests, Africa, and Australia are different than South America, the simulated NO2 and CO columns could have been different for boreal forests, Africa, Indonesian, and Australian fires had the WRF-Chem domains been set-up over each region representing a different biomass burning characteristics because meteorology and chemistry over each biomass burning region is different from South America. I suggest the authors to include a discussion on this aspect.

To clarify, we did WRF-CHEM experiments first and foremost to demonstrate the effectiveness of the Statistical Bulk Method (SBM) of sampling CO and $NO_2$ data relative to a background. In these experiments we used by design the same meteorological and chemical conditions to focus mainly on our ability to retrieve different types of burns using SBM technique. We do acknowledge that variability in meteorology and chemistry can play a significant role to CO, and in particular, $NO_2$ concentrations. For instance, additional sources of CO can come from oxidation of volatile organic compounds (VOC) and methane and removal of CO can come from oxidation with hydroxyl (OH) radicals. Dekker et al. (2019) found that these combined processes had in the end very little impact on CO concentrations. The chemistry of NOx is complex as well, as the cycling between NO and $NO_2$ takes place within minutes to hours. During the daytime there is a photochemical balance between $NO_2$ photolysis and NO oxidation by ozone converting NO into $NO_2$. The principal sink of NOx is the oxidation to $HNO_3$ in reaction with OH.

To better investigate whether how meteorology and different chemistry regimes would impact CO and $NO_2$ we sampled other regions within the South American WRF domain. In Fig. 1 here below we show for each type of burn the daily and monthly average mole density ratio (MDR) between $NO_2$ and CO sampled from WRF-CHEM and TROPOMI. For each burn we show four different WRF-CHEM MDR estimates:

(1) standard simulation with sampling at the default region (as in the paper),

(2) simulation with sampling at region 1 with every day a repeat of the same meteorology,

(3) standard simulation with sampling at alternative region 2,

(4) simulation with sampling at alternative region 2 with every day a repeat of the same meteorology.

The different chemistry regimes are mimicked by sampling in an alternative region more east in the WRF-CHEM domain, closer to the Atlantic Ocean. Other meteorological regimes are mimicked by applying constant meteorology as an alternative simulation.

As shown in Fig. 1, both meteorology and chemistry do play a role in affecting the MDR estimate. For burns where $NO_2$ emissions are relatively more prevalent, like for deforestation and savanna fires, the differences are naturally exacerbated. $NO_2$ is more affected by the chemical conditions and meteorology than CO. This is expected and is also observed in the TROPOMI data in Fig. 6a of the new manuscript (Fig. 11a of the original manuscript), where we have quite some variability in MDR for the savanna and deforestation fires across the different regions. Across the ensemble of four simulations we do find on average the same MDR signatures as detected by TROPOMI. To accommodate the concern of the reviewer, we acknowledge this in the text in Discussion Section 4, starting at line number 706: *"In general, a large part of the biases in $\Delta XNO_2$ (and thus in MDR), either caused by the sampling techniques or the instrument precision and sensitivity, were in all likelihood somewhat similar in magnitude in the regions we studied. Hence, we believe it did not impair the detection of differences in fire characteristics. The uncertainty related to chemistry and transport may have played a larger role region-to-region as it affected tropospheric $NO_2$ more differently than CO, and thus our ability to derive a robust MDR. In particular, on shorter day-to-day time scales the MDR estimates can vary greatly. The amount of OH radicals in the atmosphere acts as the primary daytime sink of $NO_2$ and can vary substantially depending on the amount of tropospheric $O_3$, water vapor and incoming sunlight (source of OH), and the presence of other chemical species such as volatile organic compounds (sink of OH). Overall, it reduces the lifetime of $NO_2$ to several hours, much shorter than the lifetime of CO. As a consequence, daily estimates of $\Delta XNO_2$ will always be biased low. In addition, daily variations in $\Delta XNO_2$ that are driven by transport and chemistry are naturally exacerbated in $\Delta XNO_2/\Delta XCO$ ratio-space. Therefore, to interpret MDR, it is currently necessary to collect multiple days of data (e.g. for an entire month) to retrieve a more robust combustion efficiency signature that cancels out some of the day-to-day variations in transport and chemistry."*

[Figure]

**Figure 1. The daily mole density ratio (MDR) between NO₂ and CO derived for four different types of burns; boreal fires, peatland fires, deforestation fires and savanna fires. For each burn four different WRF-CHEM experiments were performed outlined along the x-axis: (1) standard simulation as described in Sect 2.4 of the manuscript, (2) simulation with constant daily meteorology, (3) standard simulation sampled at the alternative region more east of the domain, (4) simulation with constant daily meteorology at the alternative region. The actual TROPOMI observed MDR based on real fires is shown by the colored symbols. The monthly average with 1σ standard deviations is shown by the diamond symbols.**

*Dekker, I. N., Houweling, S., Pandey, S., Krol, M., Röckmann, T., Borsdorff, T., Landgraf, J., and Aben, I.: What caused the extreme CO concentrations during the 2017 high-pollution episode in India?, Atmos. Chem. Phys., 19, 3433–3445, https://doi.org/10.5194/acp-19-3433-2019, 2019.*

While it was interesting to learn about TROPOMI's ability to distinguish between different biomass burning characteristics, I felt the paper should also have included a discussion on the crop-residue burning. Is it difficult to perform a similar analysis for crop-residue burning (e.g., in China or northern India) because of the limited sensitivity of TROPOMI NO2 retrievals to PBL?

We agree with the reviewer that it would have been good to include agricultural burning. In fact, we have looked into this and had hoped to include results in the manuscript, but we chose not to include this type of burning in the current study for two main reasons:

1) When focusing on large-scale agricultural regions such as in Indo-Gangetic Plain (IGP) in India we indeed observed a strong buildup of CO and $NO_2$ during the pre-monsoon wheat burning season (April-May) and post-monsoon rice burning season (October-November). However, a recent study by Dekker et al. (2019) showed that residential and commercial combustion in the same area was a much larger source of pollution than crop burning. The pollution was not only limited to New Delhi, but the accumulation of pollution extended over the entire IGP due to the meteorological conditions. This makes the differentiation between combustion of crops and other anthropogenic sources more difficult and so the ratio between $NO_2$ and CO is more likely to be misinterpreted.

2) The burning of crops is more difficult to interpret from space because the burning practices can vary quite substantially between farmers. A recent survey study by Liu et al. (under review) showed that there is a variety of burning practices and methods used in IGP (and other regions), resulting in either complete or partial burning of the crop residue. This is in particular important because it affects trace gas measurements from these burns in several ways. For instance, partial burns are less likely to be observed from space than complete burns. Partial burns also tend to release more particular matter and CO than complete burns due to smoldering combustion of wetter residues.

We therefore feel a dedicated paper focusing on the various forms of agricultural waste burning with a more regional focus would more informative and better justify the complexity of this fire type.

*Liu, T., Mickley, L.J., Singh, S., Jain, M., DeFries, R.S., and Marlier, M.E.: Crop residue burning practices across north India inferred from household survey data: bridging gaps in satellite observations, EarthArXiv, under review, 2020.*

Small Comments:

Section 2.4: Are the fire emissions subjected to plume rise in WRF-Chem?

We did not use the intrinsic plume rise calculations from WRF-CHEM. Instead, we applied a gridded characterization of measured injection height profiles for South America based on remotely sensed stereo-height information from smoke plumes (Martin et al., 2018). This allowed the trace gases from fires being emitted more realistically at multiple vertical levels in the planetary boundary layer instead just at the surface. This information was omitted by mistake in the original manuscript. In the new manuscript we explicitly mention this. In Section 2.4, at line number 403, page 20 we now write *"In addition, we used a spatial characterization of injection height profiles based on space-based stereo-height information from smoke plumes (Martin et al., 2018)"*.

*Martin, M. V., Kahn, R. A., and Tosca, M. G.: A global analysis of wildfire smoke injection heights derived from space-based multi-angle imaging, Remote Sens., 10, 1609; doi:10.3390/rs10101609, 2018.*

Line 432: Based on legends of Figure 5, I think 2.77 should be replaced with 2.97.

Correct, we changed that in the new manuscript

Line 508: Change "EF in GFED4s" to "EF in neither GFED4s"

Changed in the new manuscript

Line 553: remove "the" before "efficient".

Changed in the new manuscript

Reviewer #2

We kindly thank the reviewer for his/her time to evaluate our manuscript, and we appreciate the positive view on the work presented. Our comments appear below in red.

Despite many information provided and careful investigation, it needs to be shortened to focus on the main findings. The lengthy descriptions of each session can distract the main points of the originality of the work.

We have realized that our original manuscript has been lengthy. We appreciate the arguments made by the reviewer to focus on the main results. We have carefully searched for redundancies and we have brought the important results more to the foreground. More specifically, the results Section 3.1 now starts with the global signatures in $NO_2/CO$ (former Fig. 11, now Fig. 6) and we compare those with the simulated ratios from WRF-CHEM simulations. Subsequently, we show how the signatures compare with the efficiency signatures imbedded in the fire inventory products, GFED4s and GFAS (former Fig 12, now Fig. 7). We also include a new Table 2, showing all the values of MDR and EFR for the different regions (see specific point below). We left out the somewhat redundant discussion of the TROPOMI signatures for the different subregions (former Section 3.2). The main results and discussion of these regions are already included in Section 3.1 and Fig. 6. In Section 3.2 (former Section 3.1), we discuss the results of the South American deforestation and savanna fires in more detail, and the validation of the different sampling techniques. All in all, this reduces the text with 1200 words, we lose one figure (original Fig. 9), and we feel this improves the flow of the paper

Specific comments

The ratio between XNO2 and XCO implies not only the information on the surface emissions but also the information of its transport especially considering the longer lifetime of CO. How do you think the column comparison can cause the uncertainties of surface emissions? The author may have to comment on this concisely.

We agree that uncertainties arise from atmospheric transport and differences in lifetime of CO and $NO_2$. We discuss this in the last paragraph of the Discussion Section 4 in the new manuscript on page 35, starting at line number 706: "*In general, a large part of the biases in $\Delta XNO_2$ (and thus in MDR), either caused by the sampling techniques or the instrument precision and sensitivity, were in all likelihood somewhat similar in magnitude in the regions we studied. Hence, we believe it did not impair the detection of differences in fire characteristics. The uncertainty related to chemistry and transport may have played a larger role region-to-region as it affected tropospheric $NO_2$ more differently than CO, and thus our ability to derive a robust MDR. In particular, on shorter day-to-day time scales the MDR estimates can vary greatly. The amount of OH radicals in the atmosphere acts as the primary daytime sink of $NO_2$ and can vary substantially depending on the amount of tropospheric $O_3$, water vapor and incoming sunlight*

*(source of OH), and the presence of other chemical species such as volatile organic compounds (sink of OH). Overall, it reduces the lifetime of $NO_2$ to several hours, much shorter than the lifetime of CO. As a consequence, daily estimates of $\Delta XNO_2$ will always be biased low. In addition, daily variations in $\Delta XNO_2$ that are driven by transport and chemistry are naturally exacerbated in $\Delta XNO_2/\Delta XCO$ ratio-space. Therefore, to interpret MDR, it is currently necessary to collect multiple days of data (e.g. for an entire month) to retrieve a more robust combustion efficiency signature that cancels out some of the day-to-day variations in transport and chemistry."*

Basically, we argue that it is necessary to collect data for an entire month opposed to a single day to retrieve more robust combustion efficiency signals. Potentially, we could translate the TROPOMI retrieved column enhancement ratios into emission ratios by accounting for $NO_2$ removal by OH similar to what is demonstrated in Lama et al. (2019). This is something we would like to explore in the future; however, it requires more detailed separate analysis for each region. This study focuses on finding differences in combustion efficiency on a global scale using TROPOMI $NO_2$ and CO data "as is" without applying additional correction from other sources.

*Lama, S., Houweling, S., Boersma, K. F., Aben, I., van der Gon, H. A. C. D., Krol, M. C., Dolman, A. J., Borsdorff, T., and Lorente, A.: Quantifying burning efficiency in Megacities using $NO_2 / CO$ ratio from the Tropospheric Monitoring Instrument (TROPOMI), Atmos. Chem. Phys. Discuss., https://doi.org/10.5194/acp-2019-1112, in review, 2019.*

Are the types of burning affected by the soil and its condition? I wonder if more factors impact on the burning conditions (XNO2 and XCO ratios).

Environmental conditions, like soil moisture, do indeed play a role in affecting the combustion efficiency. For instance, combustion of wet organic matter tends to be less complete producing relatively more CO than $NO_2$ and $CO_2$. This is probably one of the reasons why we see quite some day-to-day variability in sampled $\Delta NO_2$ and $\Delta CO$ and their ratio in Fig. 6a of the new manuscript (old Fig. 11a). We can only distinguish deforestation fires from savanna fires at the peak of the Amazonian fire season, in September. The monthly average mole density ratio (MDR) is then significantly different for the two fire types. The emission factors for $NO_2$ and CO that are used by the fire emission inventories are based on a large number of different field and laboratory experiments. The average EF values that are listed in Table 1 of the manuscript do not necessarily reflect these natural variations of burning conditions. In the Introduction, at line number 70, we make the argument that the actual EFs could be different from the biome averaged values used by the fire models. Variations in the chemical and structural composition of biomass, temperature, moisture content, and wind speed can all affect combustion efficiency. Our paper aimed to understand the broad variability and we necessarily averaged over larger regions and longer timescales (see previous comment) to average out some of the variability that may be introduced on finer scales but which requires more careful accounting for transport and chemistry, amongst others.

I think the information in Figure 11 and 12 is better in the table. The table of EFs would be useful with the regions, types of burning, and seasons. That would be useful for scientific communities.

We thank the reviewer for this suggestion. We removed the barplot that is part of the old Fig. 12 (now Fig. 7) because it adds no new information and instead we inserted Table 2, which provides an overview of all MDR and EFR values derived from TROPOMI data and WRF-CHEM.

Are the ratios of deforestation fires different from all types of vegetation fires? Then it means we can capture the deforestation by MDR from space? Please describe the meaning of identifying the deforestation by the satellite sensing.

From the burns investigated, we do find that the bulk of deforestation fires in South America are clearly less efficient than savanna fires, but more efficient than peat and boreal fires. It does indicate a greater contribution from smoldering combustion of organic soils and woody debris that is typically piled together at the surface. While promising, the regional burning characteristics are currently detected at a limited length scale of 500-1000 km and at a time scale of a month. The day-to-day variability in MDR remains quite large due to natural variations of individual burns and variations in meteorology and chemistry to accurately retrieve daily combustion efficiency signals. That is why it is currently not possible with this method to pinpoint deforestation from illegal logging or mining activities at a more local level.

**Next page: Marked up manuscript with all changes that are discussed in Review rebuttals #1 and #2**

**Additional changes:**

1. **Slightly different MDR values retrieved from the WRF-CHEM simulation after we found a bug in the conversion from ppm to mol/m$^2$. The values are still within 92% reported in the original manuscript. The new values appear in Table 2, Fig. 6b and Fig. 7. It does not change the original findings of simulating clear differences in biomass burning behavior.**

2. **Number of references were missing in the reference list in the original manuscript. This has been fixed. Likewise, the original reference list included a few papers that were no longer mentioned in the main text. This also has been fixed.**

[revised manuscript text omitted]

**(a) GFED4s Emission Factor Ratio**

[Figure]

**(b) GFAS Emission Factor Ratio**

[Figure]

[Figure]

**Figure 1: Emission factor ratio between NOx and CO (EFR) for GFED4s (a) and GFAS (b). In panel (a) the range of EFR values are subdivided into three aggregated fire type categories: (1) peat and boreal fires (PEAT/BOR: <10), (2) temperate forest, deforestation and agricultural fires (TEMP/DEFOR/AGRI: between 10 and 50) and (3) savanna fires (SAV: >50). In panel (b) the range of EFR values subdivided along a different classification (see Sect. 2.1): (1) peat fires (PEAT: <10), (2) extratropical forest, tropical deforestation and agricultural fires (ETF/DEFOR/AGRI: between 10 and 30), and (3) savanna fires (SAV: >30). Regions of interest are highlighted by the red boxes.**

[Figure]

[Figure]

**Figure 2: Maps of monthly average XCO [mol m⁻²], XNO₂ [mmol m⁻²], and GFED4s CO emissions [GgCO grid⁻¹ month⁻¹] for South America (panels a-c), for Northern Africa (panels d-f), and for Southern Africa (panels g-i). Maps of daily average XCO, XNO₂ and GFED4s CO emissions are shown for Australia for December 23, 2018 (panels j-l). Regions of interest are highlighted by the red boxes.**

¶
*<object>*

[Figure]

[Figure]

**Figure 3: Three types of the statistical bulk method (SBM) are applied to an idealized distribution of XCO samples [mol m⁻²]. In panel (a) the standard SBM where ΔXCO (XCO$_{fire}$ − XCO$_{BG}$) is equal to 2σ$_{XCO}$ around the distribution mean μ$_{XCO}$. In panel (b) the first alternative SBM (SBM_alt1) where ΔXCO is equal to the difference between the 84.1 percentile rank and the 15.9 percentile rank around the median M$_{XCO}$. The estimates for ΔXCO from SBM and SBM_alt1 are only equal if the distributions are perfectly Gaussian. In panel (c) the second alternative SBM (SBM_alt2) where ΔXCO is equal to the difference between XCO$_{fire}$ from the standard SBM and XCO$_{BG}$ derived from the mean concentration of another XCO distribution sampled upwind of the fire region (dashed distribution). The estimates for ΔXCO from SBM and SBM_alt2 are only equal if XCO$_{BG}$ are the same.**

**Sample regions, WRF-CHEM domain, and PBL wind field**

[Figure]

**Sample regions, WRF-CHEM domain, and PBL wind field**

[Figure]

Figure 4: Location of the deforestation 5°×5° sampling region (green), the savanna 5°×5° sampling region (blue) and two background regions adjacent to the two source regions (purple). The WRF-CHEM domain and the predominant wind direction in the PBL during the 2018 fire season are superimposed. The location of the green region is also used to sample XCO and XNO₂ data from four WRF-CHEM simulations (see Sect 2.4).

[Figure]

**(a) XCO [mol m$^{-2}$]**
**2018-12-23**

**(b) XNO$_2$ [mmol m$^{-2}$]**
**2018-12-23**

Figure 5: The column density XCO [mol m$^{-2}$] in panel (a) and XNO$_2$ [mmol m$^{-2}$] in panel (b) measured with TROPOMI on December 23, 2018 over central Australia. The background region is depicted by the purple frame and is located upwind of multiple fire plumes. The locations of the GFED4s/GFAS fire emissions are depicted by the magenta plus-signs.

Moved (insertion) [17]

[Figure]

**(a) Global MDR from TROPOMI**

**MDR:**
- ■ 3.58 ± 1.13 (S. America savanna fires)
- ● 6.20 ± 3.78 (Australia savanna fires)
- ▼ 4.15 ± 1.07 (N. Africa savanna fires)
- ◆ 4.33 ± 1.60 (S. Africa savanna fires)
- ■ 1.55 ± 0.44 (S. America deforestation fires 1)
- ● 1.90 ± 0.94 (S. America deforestation fires 2)
- ▼ 1.93 ± 0.52 (S. America deforestation fires 3)
- ◆ 2.47 ± 0.84 (S. America deforestation fires 4)
- ■ 0.94 ± 0.37 (Sumatra peatland fires)
- ● 1.43 ± 0.92 (Borneo peatland fires)
- ■ 0.73 ± 0.20 (N. America boreal fires)
- ● 0.48 ± 0.18 (Siberia boreal fires)

**(b) Global MDR from WRF-CHEM**

**MDR:**
- ● 4.47 ± 3.67 (Savanna fires)
- ● 2.26 ± 1.74 (Deforestation fires)
- ● 1.38 ± 1.27 (Peatland fires)
- ● 0.95 ± 1.05 (Boreal fires)

[revised manuscript text omitted]

**Page 37: [1] Deleted**       **Ivar van der Velde**       **26/08/2020 19:12:00**

**Page 37: [2] Deleted**       **Ivar van der Velde**       **26/08/2020 19:12:00**

**Page 37: [3] Deleted**       **Ivar van der Velde**       **26/08/2020 19:12:00**

**Page 66: [4] Deleted**       **Ivar van der Velde**       **26/08/2020 19:12:00**

**Page 71: [5] Deleted**       **Ivar van der Velde**       **26/08/2020 19:12:00**

**Page 71: [6] Deleted**       **Ivar van der Velde**       **26/08/2020 19:12:00**